# AB Toxins as High-Affinity Ligands for Cell Targeting in Cancer Therapy

**DOI:** 10.3390/ijms241311227

**Published:** 2023-07-07

**Authors:** Ana Márquez-López, Mónica L. Fanarraga

**Affiliations:** 1The Nanomedicine Group, Institute Valdecilla-IDIVAL, 39011 Santander, Spain; amarquez@idival.org; 2Molecular Biology Department, Faculty of Medicine, Universidad de Cantabria, 39011 Santander, Spain

**Keywords:** targeted therapies, bacterial AB toxins, receptors, high-affinity, coated pits, neovasculature, translocation, drug delivery, imaging agents

## Abstract

Conventional targeted therapies for the treatment of cancer have limitations, including the development of acquired resistance. However, novel alternatives have emerged in the form of targeted therapies based on AB toxins. These biotoxins are a diverse group of highly poisonous molecules that show a nanomolar affinity for their target cell receptors, making them an invaluable source of ligands for biomedical applications. Bacterial AB toxins, in particular, are modular proteins that can be genetically engineered to develop high-affinity therapeutic compounds. These toxins consist of two distinct domains: a catalytically active domain and an innocuous domain that acts as a ligand, directing the catalytic domain to the target cells. Interestingly, many tumor cells show receptors on the surface that are recognized by AB toxins, making these high-affinity proteins promising tools for developing new methods for targeting anticancer therapies. Here we describe the structure and mechanisms of action of Diphtheria (Dtx), Anthrax (Atx), Shiga (Stx), and Cholera (Ctx) toxins, and review the potential uses of AB toxins in cancer therapy. We also discuss the main advances in this field, some successful results, and, finally, the possible development of innovative and precise applications in oncology based on engineered recombinant AB toxins.

## 1. Introduction

Cancer is a worldwide health issue with complex pathophysiology and high mortality rates. Conventional therapies, including chemotherapy, radiation therapy, and surgery, are widely used to eliminate cancer cells and increase the survival rates of patients. However, they encounter several limitations, such as cytotoxicity, multi-drug resistance, and poor selectivity since most therapies cannot distinguish between cancer and normal cells. This lack of targeting results in poor drug accumulation in the tumor and low therapeutic efficacy. Therefore, it is necessary to search for new therapies that can specifically deliver the therapeutic agent to the tumors in a way that reduces adverse effects and significantly improves efficacy [1,2].

In the search for more accurate therapies, molecular-targeted therapies have emerged as an alternative to conventional therapies as they can deliver the therapeutic agent to the tumors in a way that reduces adverse effects and significantly improves efficacy [1,2]. They can be used alone or in combination with standard chemotherapy agents [3]. These therapies include the use of small molecules with relatively low molecular weight, monoclonal antibodies, or immunotherapeutic cancer vaccines. They block the growth and spread of tumors through drugs or other substances that target specific molecules in the tumor cells or the tumor microenvironment (cancer-associated fibroblasts, immune cells, cancer stem cells, and vascular endothelial cells) [3,4]. Typical targets include cell surface antigens, growth factors, receptors, or signal transduction pathways, which, although often mutated or overexpressed in tumors, may also be present in normal tissues [3]. Although these therapies have shown successful results in the treatment of cancer, clinical resistance to these agents is still a major issue. For example, mutations of the monoclonal antibody target and downstream signaling molecules usually lead to acquired resistance due to the activation of alternative growth or survival signaling pathways [4]. Thus, it is essential to search for new ligands and target receptors for improved molecular-targeted cancer therapy.

In this field, some biotoxins have emerged as novel opportunities. Biotoxins are highly efficient natural ligands that arise in almost all forms of life, mainly for defensive purposes against predators [5,6,7]. This term includes a heterogeneous group of extremely poisonous biomolecules (small molecular compounds, proteins, and peptides) produced by the metabolism of numerous living organisms, including bacteria (bacterial toxins), fungi (mycotoxins), plants (phytotoxins), and vertebrate and invertebrate animals (zootoxins) [5,6,8]. These molecules interfere with and disrupt the physiological processes of competing organisms, and, depending on their chemical nature, they exert dose-dependent pathophysiological injury when inhaled, ingested, or absorbed [5,6,9]. One of the most appealing aspects of biotoxins is that they trigger really quick responses due to their high specificity and affinity for their receptors or intracellular molecular targets [10], being lethal at very low doses (LD_50_ < 25 mg/kg) [11]. Some of these receptors are described as being overexpressed by both, tumor cells and endothelial cells of the neovasculature of solid tumors [11,12,13,14,15,16]. Their inherent specificity and affinity for the recognition of their targets make them an invaluable source of bioactive ligands for use as pharmacological or therapeutic tools for molecularly targeted therapies. Apart from that, biotoxins are also useful tools for studying the physiological role of their molecular targets (e.g., ion channels, receptors, etc.), understanding and treating human diseases, destroying disease vectors, or treating microbial and parasitic infections [6,7,9,14]. Depending on the target tissue, biotoxins can be classified as (i) cytotoxins, such as Ricin toxin (Rtx) (Figure 1a); (ii) enterotoxins, such as Staphylococcal enterotoxins (SEs) (Figure 1b); or (iii) neurotoxins, including Batrachotoxin, Botulinum (BoNTs), and Tetanus neurotoxins (TeNT), the most potent toxins known worldwide (Figure 1c) [6,17,18,19,20,21,22,23,24].

In recent years, significant focus has been directed toward a class of biotoxins known as AB toxins. These toxins, predominantly synthesized by bacteria, have garnered considerable interest. However, it is noteworthy that AB toxins, like the Ricin toxin, can also originate from other sources, including plants. AB toxins are characterized by their unique protein structure, which consists of two distinct domains that serve different functions. Through evolutionary processes, these toxins have evolved into highly efficient carriers capable of transporting the toxic subunit into cells. This transport is facilitated by the binding of the non-toxic ligand subunit to specific cell surface receptors [25]. The molecular recognition and targeted invasion mechanisms employed by AB toxins offer valuable insights for the development of precise therapeutic approaches. The receptor-binding domain, known as the B moiety, plays a crucial role in coordinating binding to host cells and facilitating the delivery of the A moiety into the host cell cytosol. Once inside, the A moiety exerts its enzymatic effects on the cellular machinery [26,27]. Most of these toxins trigger major human disorders, usually gastrointestinal symptoms, and cause the deaths of millions of people worldwide each year [28]. However, emerging research has revealed that, when appropriately modified, AB toxins exhibit promising characteristics as adjuvants for immune stimulation, autoimmunity suppression [29], and a wide range of biomedical applications [7,14,26,29,30,31]. Furthermore, their potential applications extend to areas such as cancer diagnosis and treatment [32,33].

Here we highlight applications that utilize the specific ligand domains of AB toxins, with a particular emphasis on the B ligand domains that possess remarkable specificity and affinity for receptors that are overexpressed in solid tumors. The focus is on exploring the potential of these ligand domains in targeting and treating solid tumors, leveraging their high specificity for enhanced therapeutic outcomes [16,29,30,31,32,33,34]. AB toxins receptors overexpressed in solid tumors reviewed here include: (i) the precursor for the Heparin-Binding Epidermal Growth Factor-like Growth Factor (pro-HB-EGF), which serves as the receptor for Diphtheria toxin (Dtx) [15,16,35]; (ii) Tumor Endothelial Marker 8 (TEM8) and Capillary Morphogenesis Gene 2 (CMG2), which act as receptors for the Anthrax toxin (Atx) [11,12,36,37,38]; (iii) globotriaosylceramide (Gb3), which serves as the receptor for the Shiga toxin (Stx) [14,39], and (iv) GM1 glycosphingolipid, which acts as the receptor for the Cholera toxin (Ctx) [13] (Table 1). We will discuss the structure of these four AB toxins and their receptors, the main entry pathways, and mechanisms of action to finally discuss how to modify their B moiety to target drugs or therapeutic agents to tumors displaying their binding receptors. This novel targeted approach to AB toxins as potential agents for the treatment of cancer may serve to overcome the limitations of conventional therapies.

## 2. Generalities about AB Toxins

### 2.1. Structural Insights of AB Toxins

AB toxins are a class of toxins that contain two distinct components: the active catalytic domain (A) and the receptor binding domain (B) (Figure 2a). The B domain is responsible for coordinating the binding of the toxin to host cells and facilitating the delivery of the A domain into the host cell cytosol. This allows the A domain to exert its enzymatic effect by modifying host proteins and causing cellular changes or intoxication. In most AB toxins, the B domain also includes a translocation domain that plays a crucial role in facilitating the delivery of the A domain into the cytosol. It often forms a pore or channel that allows the A domain to traverse the membrane barrier [26,27].

Interestingly, most AB toxins are initially synthesized as inactive toxins and require proteolytic processing to convert them into their active form. The A and B moieties can be synthesized as a single polypeptide chain or independently, depending on the monomeric or oligomeric state of the B domain [27]. In single-chain AB toxins, the AB stoichiometry is 1:1, and the toxin is produced as a single polypeptide chain. Both subunits, linked by an interchain disulfide bond (Figure 2b) in the active form, are the result of proteolysis of the original polypeptide chain. However, this binding is reduced during the endo-lysosomal entry pathway, eventually releasing the catalytic domain into the cytosol of host cells. Examples of these toxins are Botulinum (BoNTs) and Tetanus (TeNT) neurotoxins, Diphtheria toxin (Dtx), Ricin toxin (Rtx), and exotoxin A [27].

In contrast, when the B domain is in an oligomeric form, the A and B moieties are produced as separate proteins that later assemble to form the holotoxin. There are two different groups of AB toxins based on the assembling process of the domains: the A + B toxins, and the AB_X_ toxins, where x is the amount of B moieties, usually 2 or 5 [26,27].

### 2.2. The Structure–Function Relationship of AB_2_-AB_5_ Toxins

The AB_2_ and AB_5_ toxins consist of two and five identical B subunits, respectively, associated with a single catalytic subunit. The holotoxin is assembled just after the synthesis of both chains. A notable feature of the catalytic A subunit of AB_5_ toxins is that, although it is a single polypeptide, after proteolytic cleavage, the active form results in two domains (A1 and A2) linked by a disulfide bond. A1 is the catalytic part itself, and A2 is an α-helix that is non-covalently inserted into the central pore that forms when the B-subunit pentamer associates [27,28] (Figure 2c). Upon reduction of the A domain intrachain disulfide bond, the catalytic domain is delivered into the cytosol [26,40,41,42,43]. AB_5_ toxins are subdivided into four different families according to their A subunit sequence homology and their catalytic activity: the (i) Cholera toxin (Ctx) family, (ii) Pertussis toxin family, (iii) Shiga toxin (Stx) family, and the (iv) Subtilase cytotoxin (SubAB) [28].

### 2.3. The Structure-Function Relationship of A + B Toxins

The A + B toxins are produced as two separate polypeptides that do not form the holotoxin complex immediately. In these toxins, the B domain needs to undergo processing by host furin proteases on the cell surface to assemble into the oligomeric form. This moiety then recruits the A moiety to form the holotoxin. The AB_7_ stoichiometry refers to the presence of seven identically processed B moieties associated with a single A subunit (Figure 2d). This is the case with the Anthrax toxin (Atx) or C2 toxin [44,45,46].

### 2.4. AB Toxins Entry Mechanisms into Cells

AB toxin entry into cell pathways relies on the B subunit’s recognition of receptors displayed on the cell surface, which can be both proteins and glycolipids. Receptor specificity is critical for the pathogenic process because it determines the susceptibility of the host, tissue tropism, and the nature and spectrum of the resultant pathology [26]. The mechanism by which AB toxins reach the cytosol is receptor-mediated endocytosis (RME). They seem to share a common mechanism of action that involves: (i) binding to specific receptors on the surface of cells displaying the receptor; (ii) internalization or translocation across the membrane; and (iii) interaction with the intracellular target [42,43]. These toxin receptors are typically distributed in lipid rafts enriched in cholesterol and glycosphingolipids [47,48]. Fluorescence microscopy studies have revealed that certain toxins, upon interaction, lead to the clustering of ligand-receptor complexes in specialized regions of the membrane called coated pits during RME. These coated pits are dependent on the presence of clathrin, a structural protein involved in vesicle formation. Pharmacological experiments have suggested that the clustering step of ligand-receptor complexes in coated pits is mediated by transglutaminase, which facilitates this clustering by generating epsilon-(gamma-glutamyl)-lysine bonds through cross-linking. This process of clustering and concentration of toxin molecules in specific regions of the cell surface is an important mechanism for the internalization of ligand-receptor complexes during endocytosis [42,49,50]. It allows for efficient uptake of the toxins by the cells, ultimately leading to their internalization and subsequent intracellular transport. Some other AB toxins enter via clathrin-independent endocytosis, through cholesterol-rich plasma membrane domains called caveolae [51].

But, despite the endocytic carrier, after receptor binding, all toxin–receptor complexes are initially delivered to early endosomal membranes [49,51], following two different pathways to translocate the A domain into the cytosol. The first pathway involves the release of the A domain from endosomes into the cytosol as a result of polypeptidic conformational changes caused by low endosomal pH. This mechanism is observed in toxins such as Dtx, botulinum toxin, tetanus neurotoxin, Atx, and C2 toxin. In contrast, certain toxins bind to membrane receptors that undergo retrograde transport, leading them to be recycled through the trans-Golgi network. This transport mechanism enables the toxins to reach the lumen of the endoplasmic reticulum (ER) via the Golgi apparatus. Once in the ER, the toxins are released into the cytosol, effectively bypassing the endo-lysosomal pathway. These include single-chain AB toxins, such as exotoxin A, ricin toxin, AB_2_ toxins, and AB_5_ toxins [28,41,51].

Based on the structure of AB toxins and the translocation pathway of the A subunit into the cytosol, these modular toxins can be categorized into four distinct groups. Toxins of group 1 include Dtx, BoNTs, and TeNT. They are single-chain AB toxins, where the A domain translocates in the endosome (Figure 3a). After a proteolytic cleavage in the peptide chain, in the active form, A and B remain linked by a disulfide bond. In the endosome membrane, the B domain forms a pore to translocate the A subunit into the cytosol after the reduction of the disulfide bond. Group 2 includes Atx and C2 toxins. They are A + B toxins, where A translocates in the endosome (Figure 3b). The heptamer-shaped ring forms a pore in the membrane of the endosome to translocate A into the cytosol. Toxins of group 3 include Ctx and Stx. They are AB_2_ or AB_5_ toxins where the A module translocates in the endoplasmic reticulum (Figure 3c). The proteolytic cleavage occurs in A, resulting in two fragments (A1 and A2) that remain linked by a disulfide bond. In the ER, A1 is translocated to the cytosol with the assistance of ER machinery (Sec proteins), which requires the reduction of the disulfide bond. Finally, group 4 includes Exotoxin A and Ricin toxin. These toxins are also single-chain AB toxins. However, A translocates in the endoplasmic reticulum (Figure 3d) [52].

Once inside the host cytosol, the A subunit exerts its cytotoxic enzymatic effect, which can be classified into three groups. The ADP-ribosyltransferase toxins catalyze the cleavage of nicotinamide dinucleotide to nicotinamide and ADP-ribose. Simultaneously, the ADP-ribose moiety is added to diphtamide, a modified histidine, arginine, asparagine, or cysteine residue, depending on the substrate specificity of each toxin. This substrate can be heterotrimeric or monomeric G proteins, cytoskeletal proteins, or the GTP-binding elongation factor 2. In this case, ADP-ribose is transferred to a modified histidine residue only present in elongation factor 2, thus inhibiting eukaryotic protein synthesis [43,51,53]. A second enzymatic effect is that triggered by the RNA *N*-glycosidases. These toxins are called RIPs and are RNA-specific *N*-glycosidases that catalyze the cleavage of a specific adenine residue (A4324) from the 28s ribosomal RNA of the large 60 s subunit. Ribosomes that contain a toxin-depurinated 28s RNA are not able to bind elongation factors, and translation is blocked. RIPs can be classified as type 1 or 2. Toxins that belong to the first group are single-chain proteins, whereas type 2 RIP toxins have A-B structure-function properties [53,54]. Finally, they can also operate as adenylate cyclases. These toxins cause a transient increase in the intracellular levels of cAMP up to 100–1000 times the physiological status. The enzymatic activity of these enzymes is stimulated by calmodulin, a Ca^2+^–binding protein present in the eukaryotic cytosol [43,53].

All of these processes have profound impacts on cellular physiology, including the inhibition of protein biosynthesis, elevation of the second messenger cAMP, disruption of the microfilament system, and suppression of neurotransmitter release. These effects collectively contribute to cell death [43].

## 3. AB Toxins That Utilize Proteins as Host Receptors

Some AB toxins exert their effects by interacting with protein receptors located on the cell surface of host cells. One such example is the precursor for the Heparin-Binding Epidermal Growth Factor-like Growth Factor (pro-HB-EGF), which is the target receptor for Dhiptheria toxin (Dtx). Additionally, Tumor Endothelial Marker 8 (TEM8) and Capillary Morphogenesis Gene 2 (CMG2) serve as receptors for Anthrax toxin (Atx). The binding of these AB toxins to their respective protein receptors initiates a cascade of cellular events leading to their toxic effects.

### 3.1. Diphtheria Toxin

#### 3.1.1. Diphtheria Toxin Structure and Mechanism of Action

Dtx is a well-studied bacterial toxin, and it was the first described example of an AB toxin. It is an ADP-ribosyltransferase produced by toxigenic strains of *Corynebacterium diphtheriae,* responsible for the symptoms of diphtheria [55]. This is a very contagious infection that affects the nose, throat, and sometimes the skin. It can be a very serious illness, and sometimes fatal symptoms include fever, sore throat, swollen glands in the neck, and a thick grey-white coating of the throat, nose, and tongue, causing difficulty breathing and swallowing. The vaccine against diphtheria was developed in 1923 [56].

Dtx is produced as a precursor form that is then cleaved of its 25 amino acid signal sequence and released into the culture medium as a 535 protein chain with a molecular weight of 58 kDa [55]. This toxin consists of three domains with different biological activities: the C-domain, or catalytic domain, which has ADP-ribosyltransferase activity (21 kDa), the T-domain, or translocation domain (20 kDa), which can cross the endosome membrane into the cytosol, and the R-domain or binding receptor domain (17 kDa) (Figure 4a). The C-domain corresponds to the A subunit of the toxin, while the T and R domains correspond to the B subunit. The A and B subunits are linked by a disulfide bond [57].

The receptor for Dtx is pro-HB-EGF (Figure 4b), which is exposed on the surface of host cells. Dtx binds to the cell surface by interacting with this receptor via the R-domain (Figure 4c), in association with CD9, a protein that does not bind directly to Dtx but increases its affinity for the receptor. Receptors are concentrated in clathrin-coated pits, and the toxin–receptor complex is internalized by clathrin-coated vesicles that are then converted into early endosomal vesicles. The clathrin triskelion is replaced with a new set of proteins, which leads to the acidification of the luminal pH of the endosomes. This acidification triggers the insertion of the T-domain into the endosomal vesicle membrane by dynamic unfolding, forming an 18–22 Å pore. This pore is essential for translocating the C-domain into the cytosol. Translocation of the A subunit is followed by reduction of the disulfide bond between the A and B subunits, resulting in the release of the C-domain into the cytosol, where it is refolded into an enzymatically active conformation. It catalyzes the NAD^+^-dependent ADP-ribosylation of elongation factor 2, inhibiting protein synthesis (Figure 4d) [55].

#### 3.1.2. HB-EGF Receptor: Physiological and Pathological Roles

HB-EGF is one of the seven ligands that bind to the epidermal growth factor receptor (EGFR/ErbB1), initially identified in the conditioned medium of human macrophage-like cells by heparin-affinity chromatography and reverse-phase liquid chromatography [14,58]. HG-EGF is a secreted peptide signaling molecule that induces the proliferation and differentiation of all cells by interacting with its receptor. It can also bind to and activate other receptors of the ErbB family of RTKs, including ErbB4, ErbB2, and ErbB3 [16,58,59].

HB-EGF is synthesized as a type I transmembrane protein called pro-HB-EGF. This membrane protein is composed of a heparin-binding domain, an EGF-like domain, a juxtamembrane domain, a transmembrane domain, and a cytoplasmic domain. Pro-HB-EGF can act in a juxtacrine manner by signaling to neighboring cells in a non-diffusible manner. Moreover, this precursor was described as the receptor for Dtx. proHB-EGF is processed by metalloproteinases in the juxtamembrane domain in a process called “ectodomain shedding.” This process yields a soluble ectodomain (sHB-EGF) and a remnant carboxy C-terminal fragment (HB-EGF-cTF). The sHB-EGF domain is a mitogen and chemoattractant for cells that express ErbB receptors. It can induce autocrine and paracrine signaling, depending on the cellular environment of the peptide. In the case of autocrine and paracrine signaling, sHB-EGF can act diffusively on cells expressing EGFR and/or ErbB4, but it can also act on the same cell that also expresses this receptor. In contrast, the HB-EGF-cTF domain acts as a signaling molecule that is phosphorylated and translocated into the nucleus, where it regulates several nuclear factors, acting in an intracrine manner [15,16,58].

HB-EGF evokes a mitogenic response in a wide variety of cell types and is expressed in a large number of tissues, suggesting a vast array of potential roles in vivo, including the regulation of heart muscle homeostasis, heart valve development, eyelid closure, wound healing, alveolarization in the lungs, and implantation of the fetus in the uterus. Dysregulation of HB-EGF shedding is implicated in several diseases, such as cardiovascular pathologies, cystic fibrosis, and various cancers, including ovarian, hepatocellular, pancreatic, gastric, breast, and colon cancers, melanoma, glioma, and glioblastoma. Several studies have demonstrated that dysregulation of expression and ectodomain shedding are related to the increased proliferative potential of these tumor cells [15,16].

In cell lines with dominant HB-EGF expression, the transfection of small interfering RNAs (siRNAs) for HB-EGF increased the number of apoptotic cells and suppressed EGFR activation. Moreover, tumor formation by human ovarian cancer cells is enhanced by the exogenous expression of pro-HB-EGF, which is blocked by siRNAs for HB-EGF and CRM197, a nontoxic mutant of Dtx. Other approaches to inhibiting and neutralizing HB-EGF are based on the use of antibodies directed against this protein. These compounds reduce the growth of glioblastoma, multiple myeloma, and some cancer cell lines, including ovarian cancer, breast cancer, and colorectal cancer. Thus, HB-EGF is considered a novel target in cancer therapy [16]. Interestingly, mice and rats do not express this receptor, so they are immune to the effects of Dtx. To study the effects of Dtx on specific cell types and disease development, Dtx-sensitive transgenic mice expressing this receptor have been developed [35].

### 3.2. Anthrax Toxin

#### 3.2.1. Antrhax Toxin Structure and Mechanism of Action

Atx is an enterotoxin produced by the Gram-positive spore-forming bacterium *Bacillus anthracis*, which is the causative agent of anthrax disease in humans and was first described by Robert Koch in 1877 [29,31,32,46,60]. This disease is contracted through spores that can reach host cells by inhalation, through the skin, ingestion, or injection, although most human diseases manifest as cutaneous anthrax (<95%). In the environment, *B. anthracis* exists as dormant spores that are very resistant to harsh environmental conditions, such as heat, radiation, pressure, chemical agents, or ultraviolet light, and can survive in the soil for decades until they are taken up by the host. Once inside a favorable environment, they germinate and grow rapidly, producing large amounts of toxins that are disseminated in the bloodstream [32,60]. The clinical symptoms range from edema and necrosis to sepsis and venous collapse. Although inhalational anthrax is not the most common disease, it shows very high mortality percentages (85–90%) compared to cutaneous anthrax (<1%) [60]. Anthrax was identified as a zoonotic disease contracted from animals and contaminated soil. However, it has been classified as a potential bioterrorism agent as it was probably used as a weapon in both the First and Second World Wars. It was also responsible for deaths in the Soviet Union in 1979, in Japan in 1955, and in the United States in 2001 [60].

Atx has a tripartite structure consisting of three independent polypeptide chains: the protective antigen (PA), the lethal factor (LF), and the edema factor (EF) (Figure 5a). PA, the B subunit, which is an 83 kDa protein (PA_83_), has the capacity of binding to target cells and escorting EF and LF from the extracellular milieu to the cytosol. LF and EF, 90 kDa and 89 kDa proteins, respectively, are the A subunits, which display different enzymatic cellular effects once they are translocated into the cytosol. PA consists of four different domains, and receptor binding occurs through some amino acid residues in domains II and IV. When PA receptor binding occurs, PA_83_ is cleaved by a furin-like protease to produce a shorter form (PA_63_) that assembles into ring-shaped heptamers that can bind up to three molecules of LF and EF [12,45,46].

Atx binds to both TEM8 and CMG2 receptors. They were described as receptors for the Atx (ANTXRs) in the early 2000s [36,37]. These are integrin-like cell surface type I transmembrane proteins, which consist of (i) an extracellular vWA domain, (ii) a metal ion-dependent adhesion site (MIDAS) motif (DXSXS…T…D, where X is any amino acid) that can bind divalent cations such as Mg^2+^ or Ca^2+^, (iii) an extracellular immunoglobulin-like (Ig-like) domain, (iv) a transmembrane domain, and (v) a long cytoplasmic tail [61,62,63]. This structure is highly conserved in mice and humans, with a 96% amino acid identity between both proteins [64]. The vWA domain is the ligand-binding domain and harbors the typical features of integrin I domains: a dinucleotide binding or Rossmann fold, with six α-helices surrounding a central six-stranded β-sheet, creating a hydrophobic core (Figure 5b). Atx binds to these receptors through hydrophobic interactions with residues of domains II and IV of PA (Figure 5c) [62]. In vitro studies have shown that the affinity of PA for TEM8 and CMG2 is similar to that of collagens for integrins [63]. Moreover, although the vWA domains of both receptors are highly conserved, the affinity of PA for CMG2 is higher, ranging from 10 to 1000-fold in vitro, which could explain why CMG2 is the main toxin receptor in mice [45,63]. While the Kd of PA for CMG2 is 170 and 780 pM (in the presence of Mg^2+^ and Ca^2+^, respectively), the Kd for TEM8 is 1.1 μM and 130 nM (in the presence of Mg^2+^ and Ca^2+^, respectively), determined by surface plasmon resonance.

TEM8 shows three different protein variants as a result of alternative splicing. The long isoform is a 564-amino-acid protein with an extracellular and transmembrane domain and a long 220-amino-acid cytoplasmic domain. The medium isoform is a 386-amino-acid protein lacking a long amino acid sequence from the cytoplasmic tail. The short isoform does not contain the transmembrane region and is predicted to be a secreted protein [12,65]. For CMG2, four different isoforms have been described. The long isoforms, CMG^488^ and CMG^489^, are almost identical apart from 12 alternative amino acids at the cytoplasmic tail. They have the same structure as that of the long isoform of TEM8. The medium isoform, CMG2^386^, lacks amino acids 213–315 of the full-length protein, corresponding to the extracellular domain, while the short isoform, CMG2^322^, is predicted to be a secreted protein [12].

In the absence of toxin molecules, TEM8 is pre-organized at the cell surface by the cortical actin cytoskeleton, but CMG2 is not. This difference is apparent when analyzing the mobility of the two receptors in the membrane using fluorescence recovery after photobleaching. TEM8 showed a lower diffusion coefficient than CMG2, and a fraction was immobile in an actin-dependent manner [50].

Upon LF and EF binding to the PA_63_ heptamer on the surface of cells overexpressing TEM8 or CGM2 receptors, these complexes are internalized into the endosomal pathway, preferentially by clathrin-mediated endocytosis, although alternative routes may exist. Recent studies have demonstrated that clathrin-mediated endocytosis of the toxin is dependent on dynamin, the heterotetrameric adaptor protein AP-1, and actin [45,63]. Upon arrival in endosomes, PA multimers transform into their pore-forming conformation [63]. Due to the low pH of endolysosomes, the pre-pore undergoes a conformational change that triggers the translocation of LF and EF into the cytosol either directly or through intraluminal vesicles (ILV). The pH threshold is lower for CMG2 than for TEM8, both in vitro (pH 5.0 for CMG2 vs. pH 6.0 for TEM8) and in vivo (pH 5.7 for CMG2 vs. pH 5.8 for TEM8) [63]. Once there, LF, a Zinc-dependent metalloprotease, cleaves members of the mitogen-activated protein kinase (MAPK) kinase family (MAPKKs or MKKs), leading to their inactivation. This results in the activation of downstream pathways including Jun-*N*-terminal kinases (JNKs), p38, and extracellular receptor kinases (ERKs), leading to necrosis and hypoxia. In contrast, EF, a calmodulin-dependent adenylyl cyclase, leads to the elevation of intracellular cAMP, resulting in edema (Figure 6) [45,46,50,66,67,68,69].

Several modifications to the cytoplasmic tails of TEM8 and CMG2 regulate the Atx endocytic pathway. The partitioning of receptors into lipid rafts depends on the palmitoylation of the tail. Palmitoylation of cytoplasmic cysteine residues increases the half-life of the receptors. Moreover, receptors are phosphorylated on tyrosine residues by Src or Fyn in response to PA binding. This posttranslational modification is not essential for the formation of the PA_63_ oligomers but impairs toxin entry. Moreover, ubiquitination occurs once PA binds to the receptor. B-arrestin2 binds to the tail and recruits an E3 ubiquitin enzyme that modifies the receptor on one or more cytosolic lysine residues [45,68].

#### 3.2.2. TEM8 and CMG2 Protein Receptors: Physiological and Pathological Roles

In humans, both TEM8 and CMG2 are mainly expressed on cells lining the entry points of this toxin: the skin, the lungs, and the small intestine [12]. Moreover, consistent with the observation that the Atx receptor is found in a variety of cell lines, TEM8 is expressed in some tissues, including the central nervous system, the heart, and lymphocytes [36]. In contrast, CMG2 mRNA expression was documented in many human tissues, including the heart, skeletal muscle, colon, spleen, kidney, liver, placenta, and lung [12].

Since the discovery of these receptors, several studies have focused on their toxin-related pathogenic roles. However, both also play physiological roles in vertebrates. The physiological roles of TEM8 and CMG2 are related to the regulation of extracellular matrix homeostasis and remodeling angiogenesis, cell migration, and cell spreading by interacting with the actin cytoskeleton, similar to integrins [5,61,70]. These functions may be related to tumor occurrence and metastasis [70]. Angiogenesis is regulated by interactions with components of the extracellular matrix (ECM). In particular, TEM8 binds to collagen I, gelatin, and the C5 domain of collagen VI [61,71,72,73,74,75]. In contrast, CMG2 selectively binds to collagen type IV and laminin [37,61]. Also, TEM8 regulates skin elasticity, proliferation, and apoptosis in chondrocytes. Mutations in TEM8 are associated with infantile hemangioma, a disease characterized by localized and rapidly growing areas of angiogenesis [12], but some are also associated with growth retardation, alopecia, false missing teeth, and optic nerve atrophy syndrome [70]. Mutations in CMG2 lead to hyaline fibromatosis syndrome [61].

TEM8 and CMG2 are both implicated in tumoral neoangiogenesis and metastatic growth [76]. TEM8 was described in the early 2000s as being highly upregulated in tumor endothelium versus normal endothelium and some tumor cells in many types of tumors, including osteosarcoma, gastric cancer, breast cancer, lung cancer, colorectal cancer, and melanoma [38,70,77]. TEM8 expression has been observed in endothelial cells during mouse embryonic development; however, in adults, it is considered specific to tumoral blood vessels, as no expression has been detected during physiological angiogenesis [11]. This is supported by several studies that have demonstrated that TEM8 knockout (KO) mice show normal angiogenesis, while murine melanoma (B16F10 cells) tumor growth is impaired in KO mice compared to wildtype (WT) mice [11,36,38,75]. These findings suggest that TEM8 is essential for tumor growth and pathological angiogenesis but not for physiological angiogenesis, making it an excellent receptor for specifically targeting cancer tissues.

CMG2 was found to be upregulated in vitro in human endothelial umbilical veins during capillary morphogenesis in three-dimensional collagen matrices [61,78], but its involvement in the process of angiogenesis in cancer is unclear in the literature. Some studies have shown that lower levels of CMG2 correlate with more aggressive soft tissue sarcoma and breast cancer, while others show that higher levels of CMG2 result in poor survival in patients with gastric cancer and glioblastoma [58].

## 4. AB Toxins That Utilize Glycosphingolipids as Host Receptors

Glycans are involved in many molecular and cellular biological processes that occur in cancer. These include cell signaling and communication, tumor cell invasion, cell-matrix interactions, neo-angiogenesis, immune modulation, and metastasis formation [79]. Changes in surface glycans are a hallmark of different cancers. The levels of glycosylation and changes in glycosyl structure are closely linked to cancer progression, malignant transformation, and drug resistance in tumors [79,80,81]. Indeed, altered glycosylation can be used as biomarkers for the early diagnosis, monitoring, and prognostication of cellular malignancy [39,82]. Here, we review globotriaosylceramide (Gb3) and GM1, receptors of Shiga toxin (Stx) and Cholera toxin (Ctx), respectively. The binding of Stx and Ctx to their receptors is influenced by the degree of unsaturation, hydroxylation, chain length, heterogeneity, and nature of the fatty acyl chains of the receptors [14,83,84].

### 4.1. Shiga Toxin

Stx is an exotoxin produced by the Gram-negative bacterium *Shigella dysenteriae* serotype 1. It was first named by the Japanese bacteriologist Dr. Kiyoshi Shiga, who first described the dysentery bacillus in 1897 [33,85]. Similar toxins are produced by some strains of *Escherichia coli* that belong to the Shiga toxin-producing *E. coli* (STEC) group. These toxins are called Stx1 and Stx2. Stx1 differs only in 1 amino acid from Stx in the A subunit, whereas the amino-acid sequence of Stx2 is only similar to the Stx sequence in 56% [33,86]. All these toxins can kill Vero cells, which is why they are also called Vero cytotoxins or verotoxins [87]. Infection with *Shigella* and toxin-producing *E. coli* can occur through the ingestion of bacteria via contaminated food or water. This infection causes bloody diarrhea that can lead to hemolytic and uremic syndrome (HUS), a fatal pediatric renal disease characterized by acute renal failure, thrombocytopenia, and anemia. This is a major public health concern that affects more than 2.8 million patients with acute infections and almost 4000 HUS cases per year. Nowadays, there are no approved treatments for preventing or treating the uremic syndrome [14,33,85].

Stx has an RNA *N*-glycosidase enzymatic A subunit (StxA, 32 kDa), composed of A1 and A2 domains linked via a disulfide bond, and five identical B subunits (StxB, 7.7 kDa) that form a pentamer that binds to the receptor. The A and B subunits are connected non-covalently by inserting the C-terminal region into the central hole of the pentamer (Figure 7a). Crystal studies have shown that there are three binding sites per B subunit, so one Stx can bind up to 15 Gb3 simultaneously (Figure 7b) [85,87,88].

The receptor for Stx is Gb3, a neutral glycosphingolipid that belongs to the group of globosides present in the extracellular leaflet of the plasma membrane. Its formula, according to the International Union of Pure and Applied Chemistry (IUPAC), is Gal-α1–4Gal-B1–4Glu-B1–Cer (Figure 7c). The B subunit of Stx binds to Gb3 through the terminal galactose disaccharide with great affinity. The Kd of StxB1 for Gb3 is 0.5–1 mM, determined by isothermal titration calorimetry [84].

Gb3 is synthesized by the addition of galactose to a lactosylceramide acceptor in a reaction catalyzed by Gb3 synthase, an α1,4-galactosyltransferase encoded by the *A4GALT* gene. It is degraded by α-galactosidase (GLA), which cleaves the α-galactose. The intra-lysosomal accumulation of undegraded Gb3 due to a deficiency of GLA leads to Fabry disease, which affects the central nervous system, heart, and kidney [84]. Gb3 was also defined as the human germinal center B-cell antigen CD77 required for B cell apoptosis, p^k^ blood group antigen expressed in red blood cells [14,84,87], and the Burkitt lymphoma antigen [87].

In humans, Gb3 is present in the renal epithelium and endothelium, microvascular endothelial cells in the lamina propria of the intestine, intestinal myofibroblasts, smooth muscle cells of the digestive tract, urogenital system, placenta, endothelial cells, dorsal root ganglion cells in the peripheral nervous system, and endothelial cells and neurons in the central nervous system [25,34,86,87]. However, the most interesting aspect of Gb3 is that it is upregulated in many human cancers including breast, ovarian, Burkitt’s lymphoma, hairy cell leukemia, megakaryoblastic leukemia, myeloma, renal, colorectal, gastric, testicular, prostate, pancreatic, sarcoma, glioma, astrocytoma, meningioma, and head and neck cancer. Interestingly, apart from the primary tumor, proliferating endothelial cells of the tumor neovasculature also express Gb3 [14,25,39,84]. Hence, Gb3 is becoming a target for cancer treatment. Gb3 has other natural ligands called lectins, proteins that can bind sugars with extremely high binding affinities. Among the most important ones are the lectin LecA from *Pseudomona aeruginosa* and members of the Stx family, previously detailed [84].

The Stx is internalized into cells by endocytosis, which involves the clustering of Gb3 receptors at the plasma membrane of target cells. This toxin can enter through both clathrin-dependent and independent pathways. In the latter case, Stx induces endocytic plasma membrane invaginations without the help of the cytosolic machinery, which results from the clustering of receptor molecules on the membrane [84,88,89]. The toxin-induced invaginations are processed by cellular machinery (dynamin, actin, and plasma membrane cholesterol) [88]. Interestingly, Stx binding to Gb3 associated with lipid rafts penetrates the cell retrogradely, sorted into the trans-Golgi network, and further into the ER, circumventing the endolysosomal vesicular pathway, thus preventing the degradation of the toxin. Then, the A subunit is processed by host furin and furin-like proteases, giving rise to polypeptidic fragments A1 and A2, which remain connected to the B subunit.

A1 and A2 are linked through a single disulfide bond that is reduced in the ER, releasing the A1 domain into the cytosol through the ER-associated protein degradation pathway to inhibit protein synthesis [43,86,87,88]. The A1 domain has *N*-glycosidase enzymatic activity, cleaving a single adenine residue from the 28s rRNA of the 60S ribosomal subunit to inhibit aminoacyl-tRNA binding and, subsequently, inhibiting host protein synthesis (Figure 8). This inhibition induces different signaling pathways in the host cell, such as the activation of ribotoxic stress and the stimulation of pro-inflammatory and pro-apoptotic signaling cascades that involve the activation of MAPKs, JNKs, p38, and ERKs [28,85,88]. This “non-canonical” receptor-mediated entry route could represent a new and interesting way to introduce nanosystems directly to the cell cytoplasm, preserving the properties of different nanosystems. This is because when nanomaterials follow the endocytic pathway, sometimes they fail to escape from the early or late endosomes, remaining entrapped. There, they face degradation in the endolysosome, so the delivery of the cargo fails [39,90,91]. Additionally, Stx binding to non-lipid raft Gb3 enters the degradative pathway to lysosomes, hence decreasing toxicity [84].

### 4.2. Cholera Toxin

The Ctx is a natural endotoxin produced by the *Vibrio cholerae* bacterium associated with aquatic environments and is the causative agent of cholera, an endemic disease in over 50 countries [92]. It can enter mammalian cells through the intestinal tract and provoke biological effects. These effects can lead to induced hemolytic and uremic syndrome, with the major symptom of watery diarrhea, and eventually to life-threatening renal failure [30].

The Ctx structure is extremely similar to that of Stx. It consists of a 28 kDa A subunit (CtxA) and a 55 kDa homopentameric CtxB subunit. CtxA is formed by two domains: toxic A1 (CtxA1) and non-toxic A2 (CtxA2). The central pore of CtxB non-covalently binds to the α-helix of CtxA2 and assembles into a heterohexameric holotoxin (Figure 7d). The difference between CtxB and StxB is that CtxB has an additional α-helix and possesses longer secondary structural elements. The five identical monomers from the CtxB subunit interact with five GM1 receptors in mammalian cells (Figure 7e) [93,94].

This toxin is a mono-sialo-glycosphingolipid belonging to the ganglioside that binds to the GM1 receptor on the cell surface [13,83,95]. GM1 forms part of the gangliotetrahexosyl series, with the formula β-Gal-(1–3)-β-GalNAc-(1–4)-[α-Neu5ac-(2–3)-]β-Gal-(1–4)-Glc-(1–1)-Cer (Figure 7f), according to IUPAC, where “Neu5ac” is sialic acid [13]. Structurally, sialic acid in humans can also be *N*-acetylneuraminic acid, *N*-Glycolylneuraminic acid, or O-acetylated sialic acid [95]. As for all gangliosides, GM1 is synthesized on the luminal membrane of the Golgi apparatus, becomes a component of released Golgi vesicles, and finally associates with the outer layer of the plasma membrane. However, a portion of the GM1 content on the membrane derives from the hydrolysis of poly-sialylated gangliosides such as GD1a, a reaction catalyzed by membrane-associated sialidase Neu3 [95]. GM1 can be further fucosylated by alpha1,2-fucosyltransferases to form fucosyl-GM1 [82].

GM1 has gained much attention since its discovery in 1963 because this ganglioside is abundant in all mammalian brains and covers 10–20% of the total ganglioside mixture [83,95]. It is normally expressed in a subset of peripheral sensory neurons and dorsal root ganglia, and it is supposed to modulate axonal outgrowth and the response of neuronal cells to signals controlling axonal extension [82]. Interestingly, it also clusters on cell surfaces where there is a conformational transition of amyloidogenic proteins, like in Alzheimer’s disease. Therefore, GM1 may be a potential therapeutic target for several intractable diseases. Fucosyl-GM1 has also been reported to be upregulated in small-cell lung carcinoma and hepatocellular carcinoma, and it has been reported to accumulate in precancerous livers of rats fed with the carcinogen *N*-2-acetylaminofluorene, so it could play a role in the development of hepatoma. However, little or no expression has been reported in other lung adenocarcinomas, bronchial carcinomas, or normal tissues, making it a novel target of small cell lung cancer and hepatocellular carcinoma [82,96].

Similar to Stx, Ctx is internalized into cells by endocytosis, which involves the clustering of GM1 receptors at the plasma membrane of target cells. These complexes are transported to endosomes and, retrogradely, to the Golgi apparatus and ER. There, the A subunit is cleaved into A1 and A2 domains, and CtxA1 enters the cytosol. It activates adenylyl cyclase to elevate intracellular levels of cAMP, thereby activating cytosolic protein kinase A (PKA). PKA-dependent pathways create an imbalance of water and electrolytes, such as by inhibiting Na^+^ absorption and stimulating the secretion of Cl^−^. Consequently, it leads to dehydration (Figure 8) [28,30].

## 5. Biomedical Applications of the AB Toxins

B toxins can be easily bioengineered for various applications. Typically, as they are bacterial proteins, the DNA sequence encoding for the A or B subunit is cloned into pET expression vectors under the control of the T7/lacO promoter and ribosomal binding site, and it is then overexpressed and purified from *E. coli* cultures.

However, for research, purified A and B moieties can also be obtained from different commercial suppliers [25].

Several approaches can be taken when considering protein toxin applications. On the one hand, the use of the holotoxin, which means the use of both the A and B subunits, enzymatic and binding receptor subunits. On the other hand, applications that only use the A or B subunits include ligand-targeted toxins, immunotoxins, suicide gene therapy, and protease-activated proteins [7].

With the idea of targeting therapies to cancer cells or the tumor microenvironment but avoiding the negative consequences of the toxic “A” subunit, in this review we focus on studies that have used the “B” subunit in different cancer therapy and diagnosis designs (Figure 9).

### 5.1. Biomedical Applications of the Diphtheria Toxin

In 1971, non-toxic or partially toxic immunologically cross-reacting forms of Dtx (CRMs) began to be isolated. A specific mutant, CRM197, was isolated in 1973 and further optimized in the mid-1980s. This mutant shares the immunological properties of the native molecule and its ability to bind the Dtx receptor [97,98] but lacks toxicity when evaluated on cellular and enzymatic assays or tested in guinea pigs [98].

This mutant displayed a great variety of applications. On the one hand, it was used as a carrier in polysaccharide or oligosaccharide conjugate vaccines, including the *Haemophilus influenzae* type b bacteria (Hib) vaccine, heptavalent and 13-valent pneumococcal vaccines, and the meningococcal serogroup C conjugate vaccine. Moreover, the CRM197-based Hib conjugate is a component of a pentavalent vaccine against diphtheria, tetanus, pertussis, hepatitis B, and Hib.

The CRM197 mutant shows a similar affinity for the Dtx receptor as the native toxin. They can bind pro-HB-EGF, as well as soluble HB-EGF, and inhibit its mitotic action by preventing its binding to ErbB receptors. In that way, both inhibitory activity toward the Dtx receptor and soluble HB-EGF and the cytotoxicity based on EF2-ADP-ribosyl activity contribute to the anti-tumorigenic effect of CRM197 [98]. This mutant was also administered intraperitoneally by Yagi et al. in an ovarian cancer murine model, producing a clear inhibition of tumorigenesis and triggering synergistic effects when administered with paclitaxel [99]. Similar results were observed in mouse models for adrenocortical carcinoma and pancreatic cancer [100,101].

Indeed, cancer patients injected with this mutant subcutaneously displayed some degree of biological anti-tumor activity [99]. Finally, CRM197 has been used for targeted drug delivery across the BBB due to the constitutive expression of HB-EGF in neurons and glial cells. This protein is highly overexpressed in Alzheimer’s disease, multiple sclerosis, stroke, epilepsy, and encephalitis. CRM197 increases BBB permeability, pinocytotic vesicles, and the redistribution of tight junction-associated proteins in brain microvasculature [100].

### 5.2. Biomedical Applications of the Anthrax Toxin

Atx has been proposed as an attractive tool for tumor therapy because the three components of the toxin are individually non-toxic, and the PA component has to be proteolytically activated before cell intake. These unique features of the toxin allow the introduction of small modifications to restrict its action to specific tumor cells [32].

The biomedical applications of Atx are mainly based on its ability to translocate different non-native proteins, drugs, and other molecules in a targeted manner to the host cell cytosol through a pore formed by PA oligomers, a property that depends on the binding of the PA to the receptor and its subsequent proteolytic activation. As a delivery system, it has similar applications to CtxB or StxB [31].

Atx PA and Atx LF have shown interesting intrinsic anti-tumor properties because they inhibit MAPKK-associated pathways, for example, in melanoma bearing the V600E BRAF mutation [102,103] and in melanoma xenografts by killing endothelial cells [104]. Similar effects were also observed in H-Ras-transformed NIH 3T3 cells [105] in soft tissue sarcomas. In these models, Atx acts directly on tumor cells, resulting in reduced growth and neo-vascularization [106]. Moreover, they proved their efficacy in a neuroblastoma mouse model, probably also due to targeting tumor vasculature [107].

Despite the efficacy of LF as a delivery system, certain cancer types have demonstrated resistance to its effects. Consequently, modifications have been made to this delivery system in order to transport enzymatic subunits of other toxins that may be more effective in targeting these resistant cells. The cellular delivery of fusion proteins or small cargo was initially reported by Arora et al. in 1992, paving the way for further advancements in the field of toxin-based therapeutic delivery systems. They designed a system formed by PA and a fusion protein (FP33) between *N*-terminal LF (LFn) and the ADP-ribosylation domain of *Pseudomonas* spp. exotoxin A (PE), which catalyzes the ADP-ribosylation of EF2 that results in protein synthesis inhibition and cell death. These experiments were performed in the CHO cell line that expresses receptors for the Atx [108]. The same group also designed new fusion proteins with LFn and StxA1 or DtxA subunits that were also successfully delivered into CHO cell cytoplasms [109]. Two years later, another group reported that this delivery system could also be used by substituting LFn for a polycationic peptide. They achieved translocation of the Lys6-DtxA fusion protein across PA pores into the CHO cell line cytosol [110]. All these delivery systems have been used by different groups to target and eliminate tumor cells in both in vitro and in vivo experiments. Different PA variants that selectively target tumor cells successfully delivered LFn-PE, resulting in tumor growth inhibition (up to 30%). This is the case with A549 tumor xenografts and non-small cell lung cancer models [111,112].

This delivery system has also been used to translocate different therapeutic cargos, such as Dox and MMAF, intracellularly. Experiments carried out in the CHO-K1 cell line show that cyclic peptides and other small molecules such as docetaxel cannot be translocated, mainly due to their large size or structural rigidity [113]. Liao et al. also employed this system to translocate antibody mimics to the cytosol of CHO cells when conjugated to LFn. In contrast, a cell-penetrating peptide was not able to deliver any of these antibody mimics [114]. Additionally, some groups have modified this delivery system to target other receptors that are not their natural receptors, such as HER2, EGFR, or the carcinoembryonic antigen, all of which are overexpressed in cancer. Finally, this system can also be employed for vaccination purposes [32].

Regarding applications of PA protein alone, there are several studies where this subunit is engineered for different approaches. For example, Rogers et al. designed a PA mutant (PA^SSSR^) that was unable to undergo normal cellular processing and remained bound to the surface receptor, inhibiting angiogenesis in vivo without endothelial cell killing [115]. Some studies also mutate PA to preferably bind to either TEM8 or CMG2 receptors. Chen et al. used a PA^N657Q^ mutation with lower cytotoxicity toward the TEM8-expressing cells, while CMG2-expressing cells were affected as with wild-type PA. However, the PA^R659S/M662R^ mutant toxin displayed enhanced specificity towards the TEM8 receptor, making this protein an interesting ligand for targeted therapies [116]. Also, PA can be mutated at its furin cleavage site to make it more specific for several cancer cell lines. For example, the furin-cleavage sequence can be changed to be cleaved by matrix metalloproteases (MMPS) and the urokinase plasminogen activator (uPA) that is overexpressed in many cancer types [31]. These AtxB applications are summarized in Table 2.

### 5.3. Biomedical Applications of the Shiga Toxin

Stx is probably the most widely applied toxin to date. It has been proposed as an attractive antineoplastic targeting tool due to several reasons. The holotoxin can induce apoptosis in various cancer cell lines in vitro in a very efficient manner, and theoretically, one single toxin molecule is enough to kill a cell [33].

Some pioneering experiments show that a single low-dose intratumoral injection of the Stx1 holotoxin was able to produce a complete regression of a 1 cm diameter human astrocytoma tumor within 10 days without reoccurrence in nude mice [117]. Furthermore, Salhia et al. demonstrated that a single intratumoral injection of Stx1 holotoxin significantly improved survival in nude mice harboring malignant meningioma intracranial tumors. Treated mice showed a reduction in tumor microvascular density and displayed increased apoptosis [118]. This group also performed experiments with ECV304 xenografts in immunocompromised mice and demonstrated that Stx1 holotoxin was enough to reduce xenograft growth and enhance mouse survival due to tumor cell cycle arrest [119]. In a posterior study, Ishitoya et al. reduced the size of human renal subcutaneous tumors in SCID mice with an intratumoral injection of Stx1 holotoxin within 5–7 days [120].

But, despite the antitumoral effects of the Stx holotoxin, its inherent toxicity and immunogenicity have prevented its use as a therapy. For that reason, it is currently preferable to employ modified toxins or use the A and B subunits separately [14,33,86]. StxB can be used by itself to specifically target Gb3-expressing cells while maintaining low immunogenicity and withstanding extreme pH conditions [33]. This subunit shows an extremely high binding affinity for membrane Gb3 (>10^−9^ M), a receptor that is widely expressed in human cancer. Besides, Stx is very resistant to hostile pH and protease conditions, both in intra- and extracellular environments, and can cross tissue barriers, facilitating their distribution within the organism. Finally, this toxin invades the target cellular cytosol by circumventing the canonical endo-lysosomal entry route, thus providing a new cell entry ligand-dependent route for lysosome-sensitive drugs [14,33]. StxB applications are summarized in Table 3.

#### 5.3.1. In Vitro and In Vivo Imaging Agents

In recent years, several StxB subunit conjugates have been developed as tools for in vitro and in vivo imaging of tumoral Gb3-expressing cells. This is useful for oncology to achieve early and specific targeting of primary tumors and metastases [14]. Janssen et al. designed coupled StxB conjugates that were used for the first time to evaluate digestive tumors. They demonstrated using confocal laser endoscopy and positron emission tomography (PET) that intravenously injected FITC-coupled StxB and 18F-labeled StxB were able to reach tumor sites in the digestive tract in a mouse model of spontaneous digestive tumorigenesis [121]. Similarly, Viel and colleagues labeled StxB with Cy5 fluorophore and demonstrated that it reached and entered the Gb3-expressing tumoral cells as well as epithelial cells of the neovascularization and monocytes and macrophages surrounding the xenografts when systemically administered to transplanted nude mice [122]. Similar studies were performed by Falguières and colleagues. They used Cy3-labeled StxB to target xenografted human colorectal tumors with high levels of Gb3, and by using fluorescence microscopy, they found that these conjugates accumulated in the tumor cells [123].

StxB has also been used as a contrast agent. Couture et al. functionalized microbubbles with StxB for ultrasonography in a breast cancer model. These bubbles were associated with cells expressing Gb3 both in vivo and in vitro. This targeted contrast agent could be used to guide therapeutic ultrasound procedures [124]. Also, gold nanoparticles can be functionalized with StxB by a non-cleavable linker and additionally functionalized with contrast agents such as gadolinium for magnetic resonance imaging (MRI) [125].

The most recent study regarding contrast agents was developed by Deville-Foillard et al. They have developed a contrast agent for MRI based on StxB. The B pentamer has been targeted with cyclic peptide scaffolds functionalized with six to nine monoamide DO3A[Gd(III)] chelates. By immunofluorescence microscopy, they demonstrated that this conjugate was accumulated in Gb3-expressing cells. To quantify the amount of conjugate accumulated in cells, they employed an inductively coupled plasma-mass spectrometry dosage of Gd(III). This system may enable the in vivo detection of tumors expressing Gb3 by MRI [126].

#### 5.3.2. Cancer Therapy

StxB has been utilized for tumor targeting and treating tumors that exhibit high levels of Gb3 expression. StxB serves as a versatile delivery system that can be engineered in various forms, such as drug conjugates, functionalized nanomaterials, and fusion proteins. By leveraging the unique binding properties of StxB to Gb3, these engineered StxB-based constructs facilitate the targeted delivery of therapeutic agents to cancer cells, offering potential benefits for tumor treatment.

In order to achieve optimal therapeutic efficacy and minimize systemic toxicity, precise control over the number of drug payloads per carrier in drug conjugates is crucial. To address this, cysteine variants with drug molecules attached to the C-terminal moiety have been developed. These variants are capable of carrying anywhere from one to five drug molecules per B monomer. This controlled approach allows for tailored drug delivery and enhances the therapeutic potential of the drug conjugates while minimizing adverse effects [25]. Kostova et al. designed StxB-drug conjugates with a cleavage linker by copper-free Huisgen [3 + 2] cycloaddition. The StxB subunit has also been linked to doxorubicin (Dox) and auristatin F (MMAF), both chemotherapies that interfere with DNA replication and tubulin polymerization, respectively. These two conjugates showed high selectivity for HT29 Gb3-positive cells, with IC_50_ values in the nanomolar range [127]. Furthermore, an StxB-SN38 prodrug was developed by conjugating camptothecin SN-38, a compound that inhibits topoisomerase I. The treatment of the Gb3-expressing gastric and pancreatic carcinoma cell line St3051 with this conjugate resulted in >100-fold increased cytotoxicity compared to irinotecan alone [128,129]. In another study, Battise et al. demonstrated that MMAE, a microtubule inhibitor used in the clinic, chemically coupled to StxB had cytotoxic properties in colorectal adenocarcinoma cells positive for Gb3 with an IC_50_ in the nanomolar range [130].

The StxB has also been linked to delivering the photosensitizers chlorine e6 and glycoporphyrin to induce reactive oxygen species by local illumination of a light-activatable photosensitizer and subsequent inflammatory responses and cell killing. StxB-e6 enhances the delivery of chlorine to cells, displaying enhanced cytotoxicity. StxB-glycoporphyrin is five times more efficient in cell cultures than the pristine molecule [131,132].

Concerning functionalized nanomaterials, we have recently shown that different nanoparticles functionalized with StxB are effectively driven into the cytoplasm of cancer cells overexpressing Gb3. Interestingly, these nanomaterials penetrate the cells using the same non-canonical receptor-mediated endocytic route described for the wild-type toxin, avoiding the endo-lysosomal pathway [90,91]. Furthermore, in in vivo studies using a preclinical mouse model of oral carcinogenesis, StxB-functionalized nanoparticles clustered on the squamous cell carcinoma lesions [90]. These nanomaterials, upon preneoplastic and malignant cancer recognition and penetration, can be used to specifically eliminate cancer cells using local photo-hyperthermia [91].

Finally, the StxB subunit has been engineered as a fusion protein to treat different types of tumors. Ryou et al. realized in vitro studies with a chimeric protein composed of Stx2B, a truncated translocation domain of StxA, and an enhanced green fluorescence protein (Stx2a^190−297-^Stx2B-EGFP). They showed that EGFP was delivered to the cytosol by furin cleavage in Vero cells and human glioblastoma, cervical, colorectal, and breast adenocarcinoma cells. Moreover, in a second study, they fused the StxB2 domain to the trans-location domain of *Pseudomonas aeruginosa* exotoxin A (TDP), and different proteins (EGFP, luciferase, adenylate cyclase) and N8A (an MDM2 inhibitor that blocks the degradation of p53 in cells) were fused to the TDP-Stx2B. All chimeric proteins were taken up by Hela and HepG2 cells. Furthermore, they showed that the N8A-TDP-Stx2B chimeric protein suppressed tumor growth in hepatic HepG2-xenografted mice when intraperitoneally injected [133,134]. Furthermore, the Dtx-StxB conjugate has been proposed as a novel therapeutic agent for antineoplastic purposes [135]. In addition, Mohseni et al. have developed a chimeric protein called Dtx^389^-StxB, which combines the B subunit of the Shiga toxin (Stx) with a mutant form of the AB subunits of Dtx. Through in vitro testing, they demonstrated that this fusion protein has promising potential as an antitumor therapy agent against breast cancer models [136]. These findings suggest the potential of these engineered constructs for targeted and effective treatment of cancer.

**Table 3 ijms-24-11227-t003:** Biomedical applications of the B subunit of the Shiga toxin. The majority of the applications of StxB are related to cancer as imaging or contrast agents, as well as agents for cancer therapy.

Type of Agent	Conjugate		Delivered Compound	Use	Model	Ref.	
Imaging	[^18^F]-Stx1B		1-[3-(2- [(18)F]Fluoropyridin- 3-yloxy)propyl]pyrrole- 2,5-dioane for PET	In vivo	Digestive cancer	[121]	
	FITC-Stx1B		Fluorescein isothiocyanate	In vivo	Digestive cancer		
	Cy5-Stx1B		Cyanine 5	In vitro, In vivo	Colorectal cancer	[122]	
	Cy3-Stx1B		Cyanine 3	In vitro	Colorectal cancer	[123]	
	6xHis:StxB-RBITC-Fe_3_O_4_@SiO_2_		Functionalized NPs	In vitro, In vivo	Head and neck cancer	[90]	
	Alexa488-StxB-Biotin-functionalized microbubbles		Ultrasound contrast microbubbles	In vitro, In vivo	Breast cancer	[124]	
	Gold StxB-functionalized NPs		Contrast agents for MRI (gadolinium, manganese, iron, etc.)	In vivo	Gb3 expressing tumors	[125]	
	StxB5- DO3A[Gd(III)]_6–9_		Contrast agents for MRI	In vitro	Gb3 expressing tumors	[126]	
Cancer therapy	Drug conjugate	Dox-Stx1B MMAF-Stx1B	Doxorubicin, MMAF	In vitro	Colorectal cancer	[127]	
		SN38-Stx1B	7-Ethyl-10- hydroxycamptothecin SN38	In vitro	Pancreatic cancer	[128]	
		MMAE-Stx1B	MMAE	In vitro	Adenocarcinoma	[130]	
	Photosensitizers conjugate	Chlorin e6-Stx1B	Chlorin e6	In vitro	Vero cells	[132]	
		TPP(p-O-beta-DGluOH) 3(p-CH2)-Stx1B	Glycoporphyrin	In vitro	HeLa cells		
	Fusion proteins	Stx2^A190−297^-Stx2B-EGFP	EGFP	In vitro	Human glioblastoma, cervical, colorectal, and breast adenocarcinoma	[134]	
		N8A-TDP-Stx2B EGFP-TDP-Stx2B	N8A-TDP	In vitro, In vivo	Liver cancer		
		DTA-StxB	DTA	In vitro	Breast cancer	[136]	
	Nanomedicine	Fe_3_O_4_@SiO_2_@RBITC@StxB:6xHis	NPs	In vitro	Head and neck cancer		[90]
		Polystyrene NPs@StxB:6xHis	Polystyrene NPs@Stx:6xHis	In vitro	Head and neck cancer		

### 5.4. Biomedical Applications of Cholera Toxin

CtxB has been widely used as a marker for caveolae and caveolar endocytosis since GM1 is partially localized to caveolae [137]. Furthermore, different types of CtxB-conjugated mesoporous silica nanoparticles have been synthesized to provide knowledge about the endocytic mechanisms, which are essential to designing cancer therapies [138,139]. These studies generate a possible application for drug delivery, especially the CtxB-conjugated mesoporous silica-supported lipid bilayer nanoparticles (protocells). These CtxB-protocells were internalized by motoneurons since GM1 is expressed in these types of cells. They remained intact in the cytosol, leading to a drug-targeting therapy [114]. Interestingly, CtxB has been conjugated with nanoparticles and other types of ligands, such as gold, nanodots, gadolinium, and saporin, for nerve tracing since GM1 expression is especially abundant in neurons. The most recent work provided a potential 2.5 nm hybrid composed of AuND-CtxB. This nanomaterial offered red fluorescence to track the sciatic nerve, besides long-term stability imaging [140].

The role of Ctx in the treatment of tumors where GM1 is overexpressed has also been reported by different groups. Ctx is used as immunotherapy to treat infections and cancers that originate in the mucous membrane, such as pancreatic or colon cancer [93]. The holotoxin has even been studied as a vaccine against human papillomavirus (HPV)-induced cervical cancer in mice, in combination with intratumoral CpG oligodeoxynucleotides [141]. Aside from being used as an inactivating toxin, CtxB offers a wide variety of applications as a fused protein and bioengineered molecule. For instance, Ji et al. engineered a recombinant CtxB vaccine that promoted immune cells against colorectal and prostate cancer and reduced the risk of death in patients compared to the cholera vaccine, although further studies were required [142,143]. Moreover, Lin et al. designed a CtxA and CtxB fusion protein composed of a macrophage-stimulating factor and a prostate-specific membrane antigen, respectively. This fusion protein showed high GM1 binding affinity, delayed prostate tumor growth, induced maturation of dendritic cells, and improved the cytotoxic T lymphocyte response [144].

Apart from vaccines, Kaur et al. demonstrated that Ctx was able to trigger growth inhibition in human small-cell lung carcinoma cell lines where GM1 is expressed. However, for non-small cell lung carcinomas, the expression of GM1 is not sufficient for this growth inhibition [145]. Moreover, Ctx has been used to treat glioma due to its ability to target GM1 receptors expressed in the blood–brain barrier (BBB), neovasculature, and glioma cells. Poly (lactic-co-glycolic acid) (PLGA) nanoparticles functionalized with CtxB could efficiently penetrate a BBB model in vitro and target glioma cells and human umbilical vascular endothelial cells. Indeed, paclitaxel-loaded PLGA particles induced apoptosis of intracranial glioma cells, and ablated the neovasculature in vivo, resulting in a significant prolongation of survival in nude mice compared to controls [146]. The same group also used CtxB-functionalized pegylated liposomes (CTB-sLip) for the diagnosis and therapy of lung metastases of colorectal cancer. Human-derived colorectal cancer cell lines demonstrated high binding affinity and cellular uptake with CTB-sLip. Moreover, these nanosystems showed elevated targeting capability for the lung metastasis of colorectal cancer in a model of nude mice in comparison to pegylated liposomes without CtxB modification [147]. CtxB applications are summarized in Table 4.

### 5.5. Issues and Challenges of Biomedical Applications of the AB Toxins

AB toxin-based therapy is a research area with great potential in different biomedical applications, including cancer therapy, as described in this section. However, the use of AB toxins as therapeutic agents encounters four main drawbacks [7].

The first is their inherent toxicity, especially when administering holotoxin. AB toxins are central to the etiology of different illnesses, including diphtheria, anthrax, cholera, and HUS. Thus, despite the antitumoral effects of holotoxins, this inherent toxicity, which relies on the A moiety, has prevented their use as therapy. For that reason, currently, it is preferable to employ modified toxins or use the B subunit separately, which maintains the high binding affinity and shows no lethal activity, lowering toxicity [14,33,86]. To address this issue, it is possible to use genetic engineering to design new proteins based only on B recognition domains to specifically target drugs or therapeutic nanomaterials to receptors that recognize these ligands.

The second is immunogenicity. As therapeutic agents based on AB toxins have at least one non-human component, antibodies against these agents can be formed and induce immune-competent patients. This results in compromised treatment efficiency by decreasing the levels of circulating functional agents. To face this issue, different approaches have been investigated, including the use of polyethylene glycol and immunosuppressive agents, deletion of B-cell and T-cell epitopes located on the surface of the toxic moieties, encapsulation of the protein, and reduction of protein accumulation by local application [13,87,148].

The third is target-independent toxicity, which means undesired toxicity in healthy tissues due to the toxin binding to cell surface components rather than specifically to its receptor. The most common toxicities are vascular leak syndrome and hepatotoxicity, which are caused by the non-specific binding of the B moiety to endothelial and hepatic cells, respectively [7]. This can be addressed by engineering B moieties with different mutations that show increased affinity for their receptors, as previously described for the Dtx and Atx applications [97,97,116].

Finally, the fourth and maybe most challenging drawback is the expression pattern of AB toxin receptors. Their expression is not always limited exclusively to the tumor site; it can be found in healthy tissues, leading to a lack of specificity and target-dependent toxicity [7,14]. Designing new therapeutic agents based on AB toxins that respond to different *stimuli* in the tumor microenvironment (i.e., nanomedicine) may be a promising approach to overcome this limitation.

## 6. Conclusions

Natural ligands have evolved over millions of years to precisely recognize specific cellular targets, ensuring their effective targeting. In the case of AB toxins, they are modular proteins that utilize their B (binding) subunit to recognize target cells, while their catalytic A (active) subunit exerts toxic and destructive effects, typically leading to cell death. Leveraging genetic engineering, it becomes possible to design novel proteins based on the B recognition domains that lack lethal activity. These engineered proteins can be employed to specifically target drugs or therapeutic nanomaterials to receptors that recognize these ligands. This approach offers a powerful strategy for the precise and targeted delivery of therapeutic agents or payloads to desired cellular receptors.

The AB toxins illustrated in this review target surface receptors expressed in tumoral cells: Stx binds Gb3, Ctx binds GM1, Atx binds both TEM8 and CMG2, and finally, Dtx recognizes pro-HB-EGF. These targets make AB toxins an attractive tool for different biomedical applications. In this review, we have focused our attention on cancer-related applications that depend on the binding of the B subunit to receptors or moieties over-expressed in different tumors, on both tumor cells and tumoral neovasculature endothelial cells.

Of all these toxins, Stx is probably the most widely used as an imaging and contrast agent with different dyes or drug molecules and as a therapeutic agent both in vitro and in vivo. Atx stands out as the second-most utilized. The reviewed studies highlight the tremendous therapeutic potential of these toxins, primarily driven by the ability of their B ligand domains to recognize target cells and facilitate the translocation of various therapeutic payloads into the cell interior. Although the applications of Dtx and Ctx in cancer therapy are not as advanced as those of other toxins, studies in different areas suggest promising future outcomes in treating various types of cancer. These findings also indicate potential avenues for developing innovative and precise applications by capitalizing on the selective advantages of AB toxins in oncology.

There are, however, several critical challenges and limitations to be addressed before the widespread use of AB toxins in oncology. These include their immunogenicity and the fact that receptor expression is not always restricted solely to the tumor site. These factors raise important questions about how to modify these ligands to enhance their systemic selectivity and overcome potential limitations in their application for cancer treatment.

## Figures and Tables

**Figure 1 ijms-24-11227-f001:**
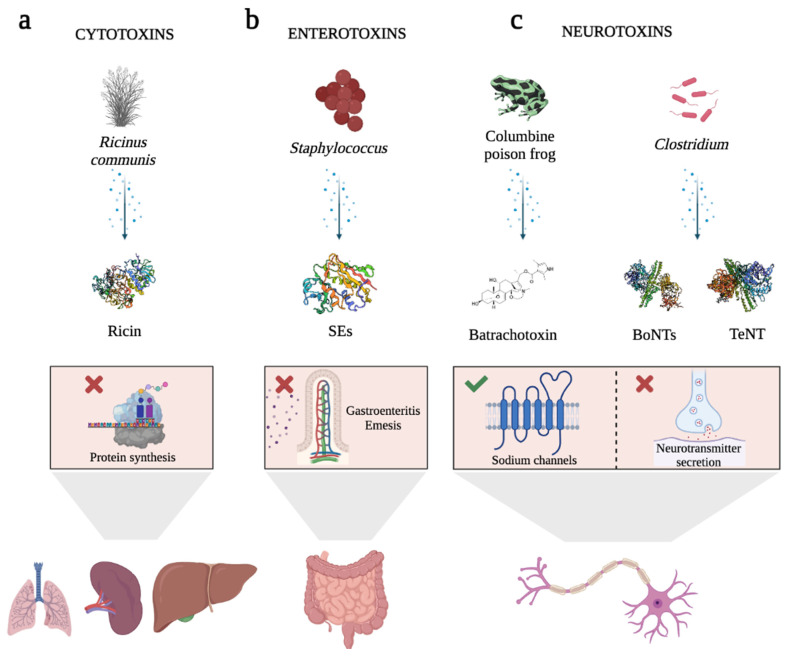
Schematic representation of the cellular effects of highly potent biotoxins. (**a**) Ricin toxin (Rtx), a protein produced by the seeds of the castor bean plant, has inhibitory effects on protein synthesis. When Rtx is ingested or inhaled, it leads to tissue damage in various organs. (**b**) Staphylococcal enterotoxins (SEs) are pyrogenic proteins that cause gastroenteritis and emesis. They induce the release of inflammatory molecules by immune system cells. (**c**) Examples of neurotoxins. Batrachotoxin, a steroid alkaloid found in the skin of the Columbine poison frog, exhibits selectivity in binding to sodium channels. This binding action causes the channels to remain open, leading to heightened permeability to sodium ions. Consequently, it induces cardiotoxicity and neurotoxicity. Among the bacterial toxins, Botulinum neurotoxins (BoNTs) and Tetanus neurotoxin (TeNT) are noteworthy examples. Produced by Clostridium species, these toxins hinder the presynaptic release of the neurotransmitter acetylcholine, thereby resulting in reversible paralysis. BoNTs primarily impede acetylcholine release in peripheral cholinergic terminals, while TeNT targets inhibitory interneurons within the spinal cord. In the graphical representation, inhibition is denoted by a red cross, while activation is indicated by a green tick. Rtx, SEs, BoNTs, and TeNT are represented as cartoon structures using Biorender software. Toxin protein structures Data Bank IDs are 2AAI for Ricin, 1SXT for SEs, 1S0F for BoNTs, and 5N0C for TeNT.

**Figure 2 ijms-24-11227-f002:**
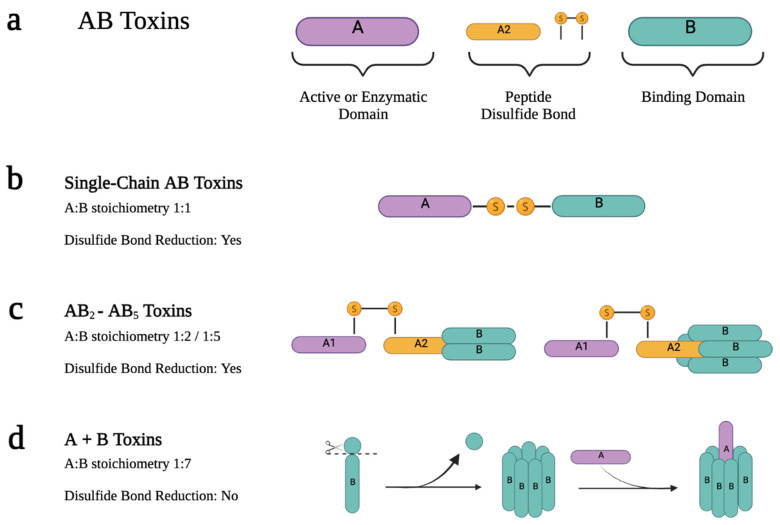
Scheme of the active forms of the different types of AB toxins. (**a)** AB toxins have two moieties: A and B. The A moiety is the active domain with enzymatic activity, while the B domain has the binding receptor property. Depending on the type of AB toxins, they can also have a linker between A and B that usually consists of a peptide and/or a disulfide bond. AB toxins active form results from a proteolytic cleavage between A and B moieties or within the A or B moiety. These toxins can be classified according to their A:B stoichiometry. (**b**) Single-chain AB toxins are produced as single polypeptide chains. They have a 1:1 A B stoichiometry, and both subunits remain linked by an interchain disulfide bond in the active form. (**c**) When the B domain is in an oligomeric state, the A and B moieties are produced as separate proteins, which assemble later on. In AB_2_ and AB_5_ toxins, A and B are assembled after the synthesis to form the holotoxin. In the active form, proteolytic cleavage occurs within the A domain, giving rise to the A1 and A2 domains that remain linked by a disulfide bond. (**d**) In A + B toxins, A and B are assembled into the de-active form after the B moiety suffers proteolytic cleavage. In this case, the holotoxin is usually in AB_7_ form.

**Figure 3 ijms-24-11227-f003:**
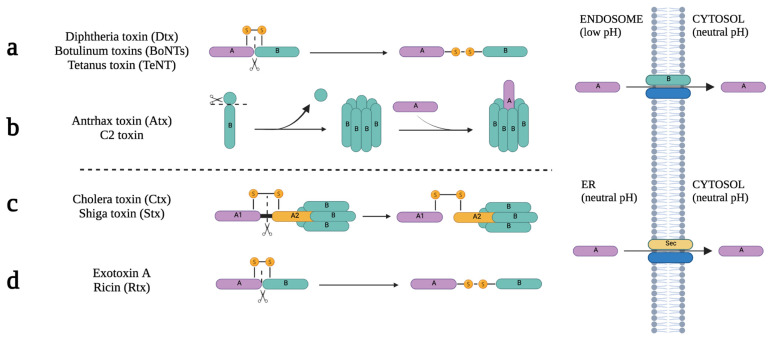
AB toxins classification, according to their structure and A translocation into the cytosol. (**a**) Toxins of group 1 (Dtx, BoNTs, TeNT) are produced as single-polypeptide precursors. These proteins are proteolytically cleaved to generate a di-chain linked by an inter-chain disulfide bond. In the endosome membrane, the B subunit forms a pore to translocate the A subunit into the cytosol. (**b**) Toxins of group 2 (Atx, C2 toxin) are produced as separated A and B proteins. The B moiety is activated by proteolytic cleavage by host furin proteases and assembles into heptamer-shaped ring structures that can bind the A moiety. B heptamers form a pore in the membrane of the endosome to translocate the A subunit into the cytosol. (**c**) Toxins of group 3 (Stx, Ctx), as in group 2, are synthesized as independent A and B proteins. However, proteolytic cleavage occurs in the A moiety, resulting in two fragments that remain linked by a disulfide bond. In the ER, A1 is translocated to the cytosol with the assistance of ER machinery (Sec proteins). This step requires the reduction of the disulfide bond. (**d**) Finally, toxins of group 4 (Rtx, Exotoxin A) are similar to toxins of group 1, but translocation of the A subunit occurs in the ER instead of the endosomes. Scissors indicate the proteolytic cleavage spot on the polypeptide toxins.

**Figure 4 ijms-24-11227-f004:**
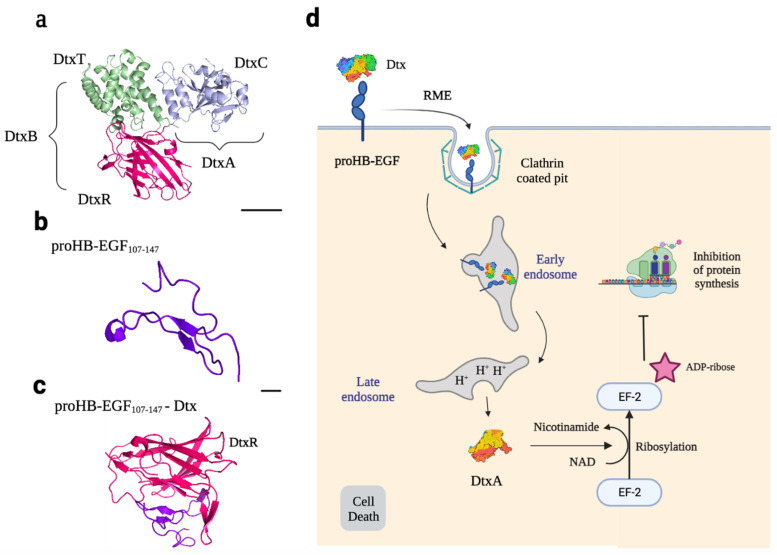
Molecular structure of Dtx and proHB-EGF, binding sites in the B subunit, and mechanism of action. (**a**) Crystallized structure of the proHB-EGF (purple) extracellular domain corresponding to residues 107–147. (**b**) Crystallized structure of Dtx. It consists of three different domains: R in pink, T in green, and C in light blue. The R and T domains are the B moiety of the toxin (binding and translocation domains), and the C domain corresponds to the A moiety. (**c**) Crystallized structure of proHB-EGF_107–147_–DtxR binding. All protein structures were depicted using PyMOL software, employing a cartoon representation. The Protein Data Bank IDs are the following: 1XDT for the extracellular HB-EGF domain, 1F0L for Dtx, and 1XDT for Dtx-HB-EGF binding. (**d**) Receptor-mediated endocytosis (RME) is the mechanism through which Dtx reaches the cytosol. It binds to the pro-HB-EGF receptor on the surface of host cells and enters the endolysosomal pathway via clathrin-coated pits. Once inside the endosome, the complex encounters an acidic environment, and the T-domain is inserted in the membrane of the endosome, forming a channel, and allowing the catalytic subunit DtxA to reach the cytosol. DtxA catalyzes NAD^+^–dependent ADP-ribosylation of EF-2, inhibiting protein synthesis. Dtx is represented as cartoon structures using Biorender software.

**Figure 5 ijms-24-11227-f005:**
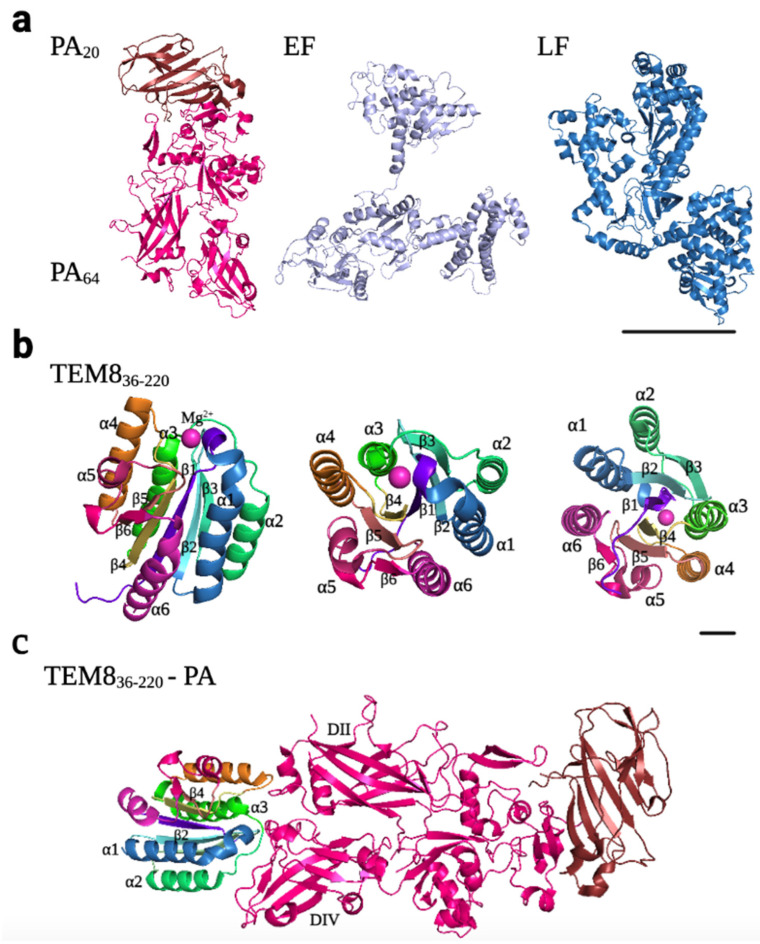
Molecular structure of Atx and TEM8 and binding sites in the B subunit. (**a**) Crystallized structure of the tripartite Atx: PA in pink and brown, EF in light purple, and LF in blue. PA is the B subunit of the toxin, and LF and EF are the catalytic subunits. (**b**) Crystallized structure of the TEM8 vWA extracellular domain corresponding to residues 36–220. This domain consists of six α-helices (α1–α6) surrounding a central hydrophobic core formed by six stranded β-sheets (β1–β6). From left to right, side, top, and bottom views of the vWA domain. Mg^2+^ ion is represented as a pink sphere bound in the MIDAS motif. (**c**) Crystallized structure of TEM_836–220_-PA Binding. TEM8 binds to Atx through hydrophobic interactions with residues of domains II and IV of PA. Residues 153–158 located in the β3–β4 loop interact with Leu_340_ and Ala_341_ of PA domain IV. Residues 113 and 115 located in the α2–α3 loop, interact with a hydrophobic cleft comprised of Leu_687_, Ile_689_, Ile_646_, Phe_678_, and Ile_656_ of PA domain IV. Residues 87 and 88 located in the β2–β3 loop interact with Asp_657_, Arg_658_, Asp_714_, and Thr_715_ of PA domain IV. Finally, residues 65 and 57 located in helix α1 interact with Tyr_688_ in PA domain II. After binding, a 20 kDa peptide (PA_20_, brown) is cut from PA_83_ by a furin protease of the host cell. PA_63_ forms a ring-shaped heptamer where EF and LF can bind. All protein structures are shown as cartoon representations using PyMOL software. The Protein Data Bank IDs are the following: 3N2N for the TEM8 extracellular domain, 1ACC for PA, 1J7N for LF, 1XFY for EF, and 1T6B for TEM8-PA binding.

**Figure 6 ijms-24-11227-f006:**
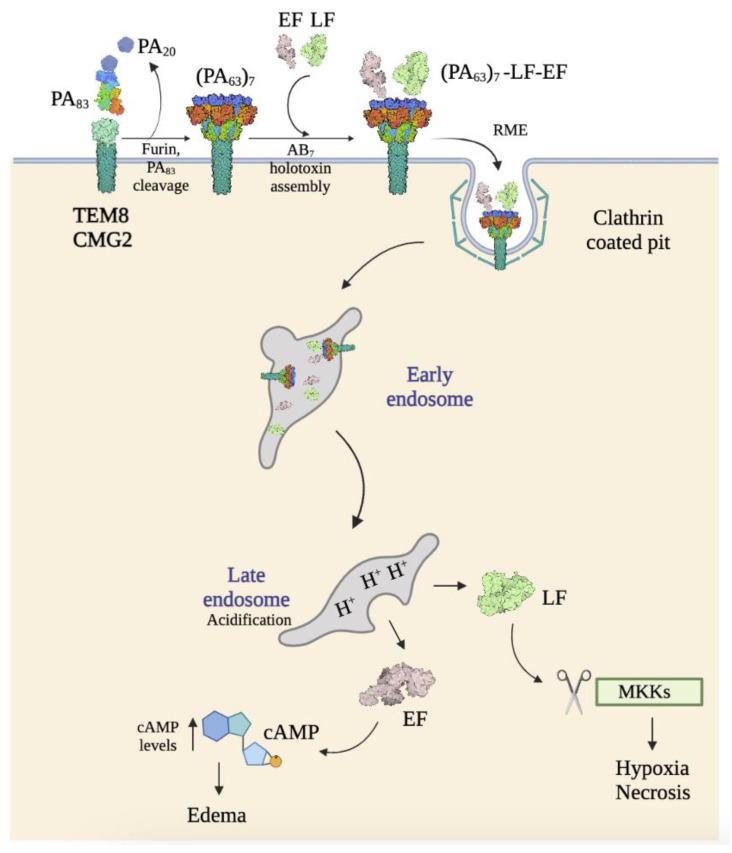
Atx entry mechanisms and cellular effects. Receptor-mediated endocytosis (RME) is also the mechanism through which Atx reaches the cytosol. It binds to TEM8/CMG2 on the surface of host cells and enters the endolysosomal pathway via clathrin-coated pits. Once inside the endosome, the complex encounters an acidic environment, and a channel is formed in the membrane of the endosome, allowing the catalytic subunits LF and EF to reach the cytosol. LF cleaves MKKs, and EF acts as a calmodulin-dependent adenylyl cyclase to increase cAMP concentrations.

**Figure 7 ijms-24-11227-f007:**
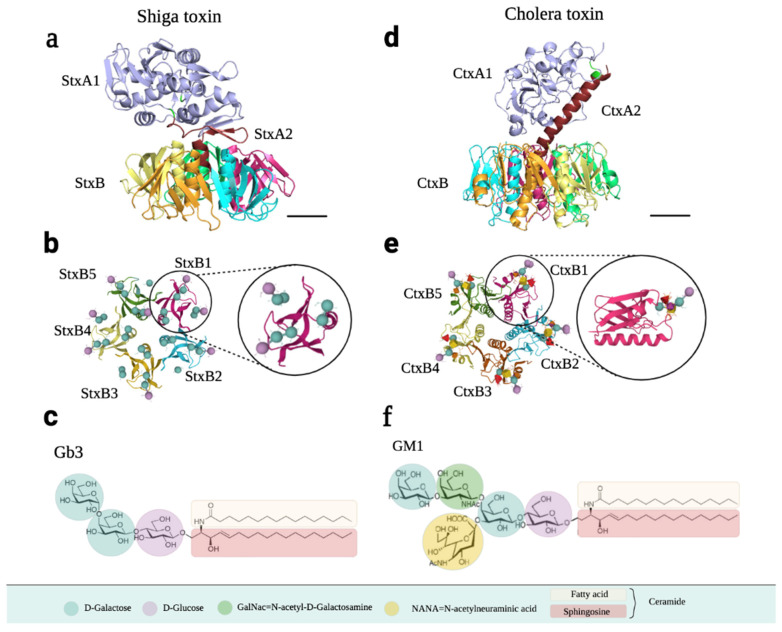
Molecular structure of Stx and Ctx and their receptors, and binding sites in the B subunit. Stx and Ctx bind glycan receptors overexpressed in some solid tumors. (**a**) Crystallized structure of the Stx. It consists of a 32 kDa A subunit (light purple and brown) and five identical 7.7 kDa B subunits (green, yellow, orange, blue, and pink) that form a pentamer. In green, the two cysteine residues form the disulfide bond that links StxA1 (light purple) and the alpha-helix StxA2 (brown). (**b**) Crystallized structure of the StxB-Gb3 binding. Each monomer of the B pentamer can bind up to three Gb3 molecules in the terminal galactose disaccharide moiety of the receptor. (**c**) Schematic representation of the Gb3 receptor structure. Gb3 is composed of a ceramide, which consists of sphingosine (light yellow) and a fatty acid molecule (light red), a molecule of D-glucose (purple), and two molecules of D-galactose (turquoise). (**d**) Crystallized structure of the Ctx. It consists of a 28 kDa A subunit (light purple and brown) and five identical 11 kDa B subunits (green, yellow, orange, blue, and pink) that form a pentamer. In green, the two cysteine residues form the disulfide bond that links CtxA1 (light purple) and the alpha-helix CtxA2 (brown). (**e**) Crystallized structure of the CtxB-GM1 binding. Each monomer of the B pentamer can only bind one GM1 molecule. (**f**) Schematic representation of the GM1 receptor. GM1, like Gb3, is composed of the ceramide group; however, the sugar moiety is different. It consists of one molecule of D-glucose, two molecules of D-galactose, one of *N*-acetylneuraminic acid (NANA, green), and one molecule of *N*-acetyl-D-galactosamine (GalNac, yellow). All protein structures were depicted using PyMOL software, employing a cartoon representation. The Protein Data Bank IDs are the following: 1R4Q for Stx, 1BOS for StxB-Gb3 binding, 1XTC for Ctx, and 5ELD for CtxB-GM1 binding.

**Figure 8 ijms-24-11227-f008:**
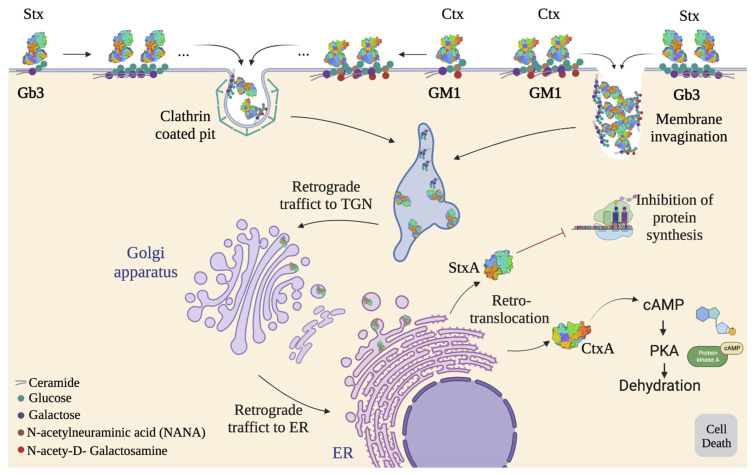
Stx and Ctx entry mechanisms and cellular effects. The mechanism by which Stx and Ctx reach the cytosol is also receptor-mediated endocytosis. They bind glycans on the surface of cells but follow a different entry mechanism than Dtx or Atx. Once they are bound to Gb3 and GM1 receptors on the surface of the host cell, they can enter via clathrin-coated pits or by invaginations of the plasma membrane. Although they are delivered to early endosomes, they undergo retrograde vesicular transport via the trans-Golgi network (TGN) and the Golgi stack to reach the ER lumen, from which the catalytic subunits StxA1 and CtxA1 are released into the cytosol. Once in the cytosol, StxA1 inhibits protein synthesis, while CtxA1 increases the amount of cAMP, leading to the dehydration of cells.

**Figure 9 ijms-24-11227-f009:**
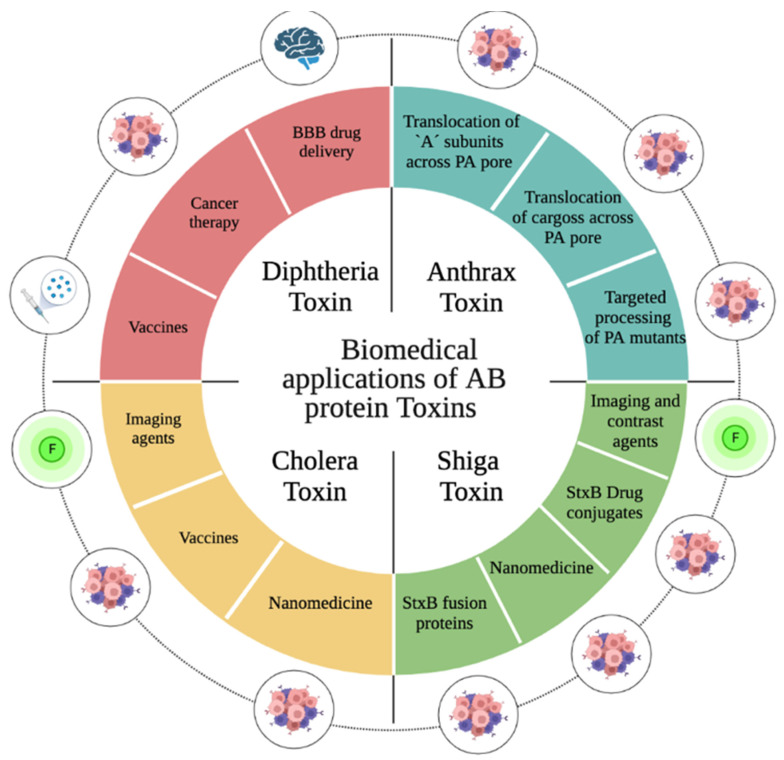
Principal biomedical applications of AB toxins. All toxins described in this review are used for cancer therapy and diagnosis, taking advantage of the B subunit binding to overexpressed receptors or as immunotoxins. However, there are more biomedical applications based on the B subunit, such as blood–brain barrier (BBB) drug delivery agents or the development of vaccines. Applications are colored in red for Diphtheria toxin, in blue for Anthrax toxin, in yellow for Cholera toxin and in green for Shiga toxin.

**Table 1 ijms-24-11227-t001:** Summary of the AB toxins. Brief description of Dtx, Atx, Stx, and Ctx structures and biological function, including information about their structure, enzymatic activity, cellular target, and cell receptor.

AB Toxin	A Subunit	B Subunit	Activity	Target	Receptor	PDB IDs
Diphtheria Toxin (Dtx)	DtxA = 21 kDa	DtxB = 37 kDa DtxBT = 20 kDa DtxBR = 17 kDa	ADP-ribosyl transferase	Elongation factor 2 (EF-2)	proHB-EGF	1F0L
Anthrax Toxin (Atx)	EF = 89 kDa LF = 90 kDa	PA = 83 kDa (×7× 63 kDa)	EF = Adenylate cyclase LF = Zn metalloprotease	Protein kinases MAPKK	TEM8 (ANTXR1) CMG2 (ANTXR2)	PA: 1ACC EF: 1XFY LF: 1J7N
Shiga Toxin (Stx)	StxA = 32 kDa StxA1 = 27.5 kDa StxA2 = 4.5 kDa	StxB = 7.7 kDa (5×)	RNA *N*-glycosidase	rRNA (28S)	Gb3 glycolipid	1R4Q
Cholera Toxin (Ctx)	CtxA = 28 kDa CtxA1 = 22.5 kDa CtxA2 = 5.5 kDa	CtxB = 11 kDa (5×)	ADP-ribosyl transferase	Adenylate cyclase	GM1 ganglioside	1XTC

**Table 2 ijms-24-11227-t002:** Biomedical applications of the B subunit of the Anthrax toxin. AtxB is mainly used along with LF or another peptide to translocate peptides or small drug molecules into the cytosol of cancer cells overexpressing TEM8 and/or CMG2 receptors.

Type of Agent	Conjugate	Delivered Compound	Use	Model	Ref.
Cancer therapy	PA-LF	LF	In vitro, In vivo	Melanoma bearing V600E BRAF mutation, soft tissue sarcomas, neuroblastoma	[102,103,104,106]
	PA LFn-PEA	PEA	In vitro	CHO cells	[108]
		PEA	In vivo	A549 tumor xenografts and non-small cell lung cancer	[112,113]
	PA LFn-Dox PA LFn-MMAF	Dox MMAF	In vitro	CHO cells	[114]
	PA LFn-StxA1 PA LEFn-DtxA	StxA1 DtxA	In vitro	CHO cells	[109]
	PA Lys6-DtxA	DtxA	In vitro	CHO cells	[111]
	PA^SSSR^		In vivo	Angiogenesis reduction	[116]
	PA^N657Q/R659S/M662R^		In vitro	CHO cells, mouse embryonic fibroblast	[97]

**Table 4 ijms-24-11227-t004:** Biomedical applications of the B subunit of the Cholera toxin. Applications of CtxB include its use as an imaging agent for neuromuscular diseases due to the expression of GM1 receptors in these types of cells. However, there are also several applications for cancer therapy, including vaccines and nanomedicine.

Type of Agent	Conjugate		Use	Model	Ref.
Imaging	CtxB-protocells		In vitro, In vivo	Neuromuscular disorders	[139]
	AuND-CtxB		In vivo	Tracking of sciatic nerve	[140]
Cancer therapy	Vaccines	Ctx-E7	In vivo	Human papilloma virus (HPV)-induced cervical cancer	[141]
		Recombinant CtxB	*Clinial Trials*	Prostate and colorectal cancer	[142,143]
		CtxA and CtxB fusion protein	In vivo	Prostate cancer	[144]
	Nanomedicine	PLGA-CtxB	In vitro, In vivo	Glioma	[146]
		CtxB-sLip	In vitro, In vivo	Lung metastasis and colorectal cancer	[147]
	Holotoxin	Ctx	In vitro	Human small cell lung carcinoma	[145]

## Data Availability

Not applicable.

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
