# Peer review of "AB Toxins as High-Affinity Ligands for Cell Targeting in Cancer Therapy"

_ijms, 2023, doi:10.3390/ijms241311227_

Round 1
Reviewer 1 Report
This review has provided lots of details on different types of AB toxins and their pathways into cells. The topic is interested and most of the contents are very helpful to readers in biomedical research fields. However, some sections need to be better organized, some sentences and figures have misleading typos and wording problems. Careful revision is required before it can be published.
1. Page 5 line 137, “…when domain the `B´ domain is in an oligomeric form…”, is this a typo?
2. Page 5 line 138 to line 139, “…two different groups of AB toxins can be distinguished, the AB2 and AB5 toxins, and the A-B toxins…”, the name A-B toxins is confusing with AB toxin, is there any better name for this type?
3. Page 5, section 2.2 and 2.3, it is better to add another figure to better understand those structures.
4. Page 5, line 177, “…But, independent of the endocytic carrier, all toxin-receptor complexes are initially delivered to early endosomes…”, the word “independent” is misleading, “despite” is better.
5. Page 6, line 188 to line 193, the contact here is very confusing and misleading, compared to the Figure 2 caption, need to be clarified.
6. Page 7, line 246, is this C-domain or A-domain?
7. Page 8, Figure 3D, a star shape is attached to EF-2, what does it mean here? Need to be clarified in Figure 3 caption.
8. Page 8, line 287 to line 289, this sentence is really confusing, how many domains this precursor has?
9. Section 3.1 has only one subsection 3.1.1? The structure of this part needs to be reorganized.
10. Page 11, Figure 4, the caption of Figure 4A should be for 4B, caption of 4B for 4A.
11. Page 13, Figure 5, lower right corner, what does MKKs means here?
12. Section 3.2 from line 340 to line 415, this part is too long, needs to be shortened. The authors should focus on mechanisms that are closely related to cancer treatments.
13. Page 15, line496, does it mean more than 2.8 million patients per year?
14. In Figure 6, caption for 6A should be for 6C, and caption for 6D should be for 6F. Other captions should also be corrected according to what they are really referring to.
15. Page 17, line 570 to 572, why does this “non-canonical” receptor-mediated entry route could represent a new and interesting way to introduce nanosystems directly to the cell cytoplasm, preserving the properties of different nanosystems? More explanation is needed here.
16. In Table 4, contrast agents are essentially imaging agents, the first column should be modified.
17. Page 23, line 770, Table 4 should be Table 3. Page 27, line 896, Table 3 should be Table 4. Context referring to these two tables should also be modified.
18. In Table 2, Table 3, and Table 4, which work is involved in clinical trial? The in vitro, in vivo, and clinical studies are very different, should be marked in those tables accordingly.
19. The authors only used two sentences at the very end of Conclusion section to discuss the issues and challenges, that is not enough. A sub-section is needed in section 5 to discuss this topic.
Please check my previous points #1, #4, and #8
Author Response
Answers Reviewer 1
First, we would like to thank the reviewers for reading and critiquing our review manuscript and for the opportunity they have given us to improve the text with their comments and suggestions. Below we respond point by point to the comments and indicate the changes we have introduced.
Comment #1:
Page 5 line 137, “…when domain the `B´ domain is in an oligomeric form…”, is this a typo?
Answer:
We thank the reviewer for this remark. The typo is now corrected.
Comment #2:
Page 5 line 138 to line 139, “…two different groups of AB toxins can be distinguished, the AB2 and AB5 toxins, and the A-B toxins…”, the name A-B toxins is confusing with AB toxin, is there any better name for this type?
Answer:
After reading this comment, we have reconsidered naming A-B toxins as A+ B toxins because the A moiety can only be assembled with the B moiety after it suffers a catalytic cleavage. This new name has been corrected on the text and figures.
Comment #3:
Page 5, section 2.2 and 2.3, it is better to add another figure to better understand those structures.
Answer:
As suggested, we have added an extra figure (new Figure 2) to show more clearly the structure of single-chain AB toxins, AB2 and AB5 toxins, and A + B toxins. As this figure has been added, the following ones have been changed in their numeration in the main text. Moreover, we have changed some sentences from 2.1, 2.2, and 2.3 sections for a better understanding of these structures. Corrected text is marked in red in pages 3 and 4.
Comment #4:
Page 5, line 177, “…But, independent of the endocytic carrier, all toxin-receptor complexes are initially delivered to early endosomes…”, the word “independent” is misleading, “despite” is better.
Answer:
Thank you for this appreciation. We have changed the text that now reads: “But, despite of the endocytic carrier, after receptor binding, all toxin-receptor complexes are initially delivered to early endosomal membranes [49,51] following two different pathways to translocate the A domain into the cytosol (Fig. 2).” In lines 181-182.
Comment #5:
Page 6, line 188 to line 193, the contact here is very confusing and misleading, compared to the Figure 2 caption, need to be clarified.
Answer:
As requested, we have added more explanation in the main text and some details on the new Figure 3 caption to make it clearer. Please see corrections in new text lines 193-208, pages 4-5, and the corresponding Figure 3 legend. Corrections are shown in red in the manuscript text.
Comment #6:
Page 7, line 246, is this C-domain or A-domain?
Answer:
The Active Domain or A domain in the Dtx is called C-domain. The C-domain corresponds to the A subunit of the toxin while the T and R domains correspond to the B subunit. A and B subunits are linked by a disulfide bond. We have corrected it on new Figure 4 and in the figure legend so it is better explained. Text corrections of the legend are marked in red.
Comment #7:
Page 8, Figure 3D, a star shape is attached to EF-2, what does it mean here? Need to be clarified in Figure 3 caption.
Answer:
Thank you for the remark. The star shape corresponds to ADP-ribose of EF-2 after ribosylation process. We have modified new Figure 4 to make it clearer.
Comment #8:
Page 8, line 287 to line 289, this sentence is really confusing, how many domains this precursor has?
Answer:
Thank you for this comment. This membrane protein (pro-HB-EGF) has 5 domains: a heparin-binding domain, an EGF-like domain, a juxtamembrane domain, a transmembrane domain, and a cytoplasmic domain. The precursor of this protein (pre-pro-HB-EGF) has two more domains (not explained on the main text). The text has been corrected accordingly (see new lines 274-276 in page 6, marked in red).
Comment #9:
Section 3.1 has only one subsection 3.1.1? The structure of this part needs to be reorganized.
Answer:
We have added two new sections, both for the Dtx and Atx. Text structure now reads:
3.1. Diphtheria toxin
3.1.1. Diphtheria toxin structure and mechanism of action
3.1.2. HB-EGF receptor: physiological and pathological roles
3.2. Anthrax toxin
3.2.1. Anthrax toxin structure and mechanism of action
3.2.2. TEM8 and CMG2 protein receptors: physiological and pathological roles
Comment #10:
Page 11, Figure 4, the caption of Figure 4A should be for 4B, caption of 4B for 4A.
Answer:
We thank the reviewer for this comment. The figure caption (now Figure 5) has now been corrected.
Comment #11:
Page 13, Figure 5, lower right corner, what does MKKs means here?
Answer:
MKKs are MAPK kinases, also known as MAPKKs. This has been corrected on the main text (new lines 383-384, page 8) and new Figure 6. We use the term MKKs instead of MAPKKs as in Figure 6.
Comment #12:
Section 3.2 from line 340 to line 415, this part is too long, and needs to be shortened. The authors should focus on mechanisms that are closely related to cancer treatments.
Answer:
In our opinion, we would not change this part of the review because this is the mechanism by which Atx binds to its receptors and the mechanism of entry into the cells and action, which is essential to develop different therapies based on this toxin. This same information is also explained for the other 3 toxins described in the review.
Comment #13:
Page 15, line 496, does it mean more than 2.8 million patients per year?
Answer:
Thank you for this remark. Yes, it means more than 2.8 million patients per year. It has been corrected on the main text (new lines 460-462 marked in red, page 10).
Comment #14:
In Figure 6, caption for 6A should be for 6C, and caption for 6D should be for 6F. Other captions should also be corrected according to what they are really referring to.
Answer:
We thank the reviewer for this comment. The figure caption (now Figure 7) has now been corrected.
Comment #15:
Page 17, line 570 to 572, why does this “non-canonical” receptor-mediated entry route could represent a new and interesting way to introduce nanosystems directly to the cell cytoplasm, preserving the properties of different nanosystems? More explanation is needed here.
Answer:
As requested, we have added a brief explanation of why this entry route is interesting for introducing nanosystems into cells (lines 519-522 page 11, marked in red).
Comment #16:
In Table 4, contrast agents are essentially imaging agents, the first column should be modified.
New Table 3. We have changed the name of the column to Imaging agents, and also in 5.3.1 section in the main text.
Comment #17:
Page 23, line 770, Table 4 should be Table 3. Page 27, line 896, Table 3 should be Table 4. Context referring to these two tables should also be modified.
Answer:
Thank you for this remark. We have already changed it also in the main text.
Comment #18:
In Table 2, Table 3, and Table 4, which work is involved in clinical trial? The in vitro, in vivo, and clinical studies are very different, should be marked in those tables accordingly.
Answer:
To our knowledge, there are only two studies from this review that are involved in clinical trials: refs: 143 and 144 (Table 4). We have already indicated it in the corresponding Table. The rest of the studies are in vitro if they refer to the work with cell lines or in vivo if they involve the use of animal models.
Comment #19:
The authors only used two sentences at the very end of Conclusion section to discuss the issues and challenges, that is not enough. A sub-section is needed in section 5 to discuss this topic.
Answer:
Considering this comment, we have added a sub-section 5.4 called Issues and challenges of biomedical applications of the AB toxins. Moreover, we have changed some paragraphs in section 6 Conclusions so the information is better organized.

Reviewer 2 Report
This manuscript deals with application of AB toxins as novel alternatives to conventional cancer therapies. The authors also introduce the current situation of AB tioxin research in the field of cancer therapy. This manuscript is well organized and scientifically sound. In addition, considering that the importance of the development of nove cancer therapies, this manuscript provides valuable information on AB toxins as alternative cancer therapies. I suggest several things to improve the quality of this manuscript.
<Minor points>
1. In Introduction, the third paragraph (In the search~of solid tumor) seems to be inappropriately located. It will seem to be more natural if the third paragraph is located after the fourth paragraph, and some phrases are modified in keeping with the overall flow between paragraphs.
2. In Table 1, the table can include PDB IDs of the respective proteins.
3. In 2.1 Structural insights of AB toxins section, a general and schematic figure of a typical AB toxin can be inserted for readers to easily understand the AB toxin structure.
4. In Conclusions, the authors can state the current situation or efforts to improve the selectivity of AB toxins in cancer therapy with some references.
5. Although the authors state that AB toxins have high affinity to cancer cells in the title, I cannot find any objective parameters to show the high affinity of AB toxins to cancer cells. Thus, the authors need to show any numerical values for the high affinity, if possible.
Minor editing of English language is required.
Author Response
Answers Reviewer 2
First, we would like to thank the reviewers for reading and critiquing our review manuscript and for the opportunity they have given us to improve the text with their comments and suggestions. Below we respond point by point to the comments and indicate the changes we have introduced.
#Comment 1:
In Introduction, the third paragraph (In the search~of solid tumor) seems to be inappropriately located. It will seem to be more natural if the third paragraph is located after the fourth paragraph, and some phrases are modified in keeping with the overall flow between paragraphs.
Answer:
Thank you for this comment. As requested, we have changed some sentences in the Introduction to make it more fluent while keeping the overall flow. The corrected text is marked in red.
#Comment 2:
In Table 1, the table can include PDB IDs of the respective proteins.
Answer:
Thank you for this appreciation. We have already included the PDBs IDs in Table 1.
#Comment 3:
In 2.1 Structural insights of AB toxins section, a general and schematic figure of a typical AB toxin can be inserted for readers to easily understand the AB toxin structure.
Answer:
As suggested, we have added an extra figure (new Figure 2) to show more clearly the structure of single-chain AB toxins, AB2 and AB5 toxins, and A + B toxins. As this figure has been added, the following ones have been changed their numeration in the main text. Moreover, we have changed some sentences from 2.1, 2.2 and 2.3 sections for a better understanding of these structures. Corrected text is marked in red in pages 3 and 4.
#Comment 4:
In Conclusions, the authors can state the current situation or efforts to improve the selectivity of AB toxins in cancer therapy with some references.
Answer:
Considering this comment, we have added a sub-section 5.4 called Issues and challenges of biomedical applications of the AB toxins. Moreover, we have changed some paragraphs in section 6 Conclusions so the information is better organized.
#Comment 5:
Although the authors state that AB toxins have a high affinity to cancer cells in the title, I cannot find any objective parameters to show the high affinity of AB toxins to cancer cells. Thus, the authors need to show any numerical values for the high affinity, if possible.
Answer:
AB toxins show very high affinities for their receptors, that are overexpressed in tumoral cells and cells of the tumoral endothelium. That is why we refer to them as high-affinity ligands in the title and the rest of the review. Thus, these ligands can be used as high-affinity ligands to specifically target cells overexpressing their receptors. It can be noticed in all the references of this review that authors have used these ligands to successfully inhibit tumor progression. As requested, we have included Kd values for StxB and AtxB in new lines 472-473 for Stx and 354-356 for Atx. The text is marked in red.

Round 2
Reviewer 1 Report
The revision has significantly improved the manuscript. It can be published after careful proof-reading.